# Optimal Weak to Strong Learning

**Kasper Green Larsen**
Department of Computer Science
Aarhus University
Aarhus, Denmark
larsen@cs.au.dk

**Martin Ritzert**
Department of Computer Science
Aarhus University
Aarhus, Denmark
ritzert@cs.au.dk

## Abstract

The classic algorithm AdaBoost allows to convert a weak learner, that is an algorithm that produces a hypothesis which is slightly better than chance, into a strong learner, achieving arbitrarily high accuracy when given enough training data. We present a new algorithm that constructs a strong learner from a weak learner but uses less training data than AdaBoost and all other weak to strong learners to achieve the same generalization bounds. A sample complexity lower bound shows that our new algorithm uses the minimum possible amount of training data and is thus optimal. Hence, this work settles the sample complexity of the classic problem of constructing a strong learner from a weak learner.

## 1 Introduction

The field of boosting has been started from a classic question in learning theory asking whether classifiers that are just slightly better than random guessing can be used to create a classifier with arbitrarily high accuracy when given enough training data. This question was initially asked by Kearns and Valiant [15, 16] and ignited the line of research that eventually lead to the development of AdaBoost [7], the prototype boosting algorithm to date. AdaBoost carefully combines the predictions of several inaccurate classifiers trained with a focus on different parts of the training data to come up with a voting classifier that performs well everywhere.

We quantify the performance of an inaccurate learner by its *advantage* $\gamma$ over random guessing. Said loosely, a $\gamma$-*weak learner* will correctly classify new data points with probability at least $1/2 + \gamma$. In contrast, given $0 < \varepsilon, \delta < 1$ and enough training data a *strong learner* outputs with probability $1 - \delta$ over the choice of the training data and possible random choices of the algorithm a hypothesis that correctly classifies new data points with probability at least $1 - \varepsilon$. The number of samples $m(\varepsilon, \delta)$ such that the learning algorithm achieves the desired accuracy and confidence levels is called the *sample complexity*. The sample complexity is the key metric for the performance of a strong learner and depends on the weak learner's advantage $\gamma$, the weak learner's flexibility measured in terms of the VC-dimension, as well as $\varepsilon$ and $\delta$. Essentially, a construction with low sample complexity makes the most out of the available training data.

AdaBoost [7] is the classic algorithm for constructing a strong learner from a $\gamma$-weak learner. If the weak learner outputs a hypothesis from a base set of hypotheses $\mathcal{H}$, then AdaBoost constructs a strong learner by taking a weighted majority vote among several hypotheses $h_1, \ldots, h_t$ from $\mathcal{H}$. Each of these hypotheses is obtained by invoking the $\gamma$-weak learning algorithm on differently weighted versions of a set of training samples $S$. The number of samples required by AdaBoost for strong learning depends both on the advantage $\gamma$ of the weak learner and the complexity of the hypothesis set $\mathcal{H}$. If we let $d$ denote the VC-dimension of $\mathcal{H}$, i.e. the cardinality of the largest set of data points $x_1, \ldots, x_d$ such that every classification of $x_1, \ldots, x_d$ can be realized by a hypothesis $h \in \mathcal{H}$, then it

36th Conference on Neural Information Processing Systems (NeurIPS 2022).

is known that AdaBoost is a strong learner, which for error $\varepsilon$ and failure probability $\delta$, requires

$$O\left(\frac{d \ln(1/(\varepsilon\gamma)) \ln(d/(\varepsilon\gamma))}{\varepsilon\gamma^2} + \frac{\ln(1/\delta)}{\varepsilon}\right), \tag{1}$$

samples. This sample complexity is state-of-the-art for producing a strong learner from a $\gamma$-weak learner. However, is this the best possible sample complexity? This is the main question we ask and answer in this work.

First, we present a new algorithm for constructing a strong learner from a weak learner and prove that it requires only

$$O\left(\frac{d}{\varepsilon\gamma^2} + \frac{\ln(1/\delta)}{\varepsilon}\right)$$

samples. In addition to improving over AdaBoost by two logarithmic factors, we complement our new algorithm by a lower bound, showing that any algorithm for converting a $\gamma$-weak learner to a strong learner requires

$$\Omega\left(\frac{d}{\varepsilon\gamma^2} + \frac{\ln(1/\delta)}{\varepsilon}\right)$$

samples. Combining these two results, we have a tight bound on the sample complexity of weak to strong learning. In the remainder of the section, we give a more formal introduction to weak and strong learning as well as present our main results and survey previous work.

## 1.1 Weak and strong learning

Consider a binary classification task in which there is an unknown concept $c : X \to \{-1, 1\}$ assigning labels to a ground set $X$. The goal is to learn or approximate $c$ to high accuracy. Formally, we assume that there is an unknown but fixed data distribution $\mathcal{D}$ over $X$. A learning algorithm then receives a set $S$ of i.i.d. samples $x_1, \ldots, x_m$ from $\mathcal{D}$ together with their labels $c(x_1), \ldots, c(x_m)$ and produces a hypothesis $h$ with $h \approx c$ based on $S$ and the labels. To measure how well $h$ approximates $c$, it is assumed that a new data point $x$ is drawn from the same unknown distribution $\mathcal{D}$, and the goal is to minimize the probability of mispredicting the label of $x$. We say that a learning algorithm is a weak learner if it satisfies the following:

**Definition 1.** *Let $C \subseteq X \to \{-1, 1\}$ be a set of concepts and $\mathcal{A}$ a learning algorithm. We say that $\mathcal{A}$ is a $\gamma$-weak learner for $C$, if there is a constant $\delta_0 < 1$ and an integer $m_0 \in \mathbb{N}$, such that for every distribution $\mathcal{D}$ over $X$ and every concept $c \in C$, when given $m_0$ i.i.d. samples $S = x_1, \ldots, x_{m_0}$ from $\mathcal{D}$ together with their labels $c(x_1), \ldots, c(x_{m_0})$, it holds with probability at least $1 - \delta_0$ over the sample $S$ and the randomness of $\mathcal{A}$, that $\mathcal{A}$ outputs a hypothesis $h : X \to \{-1, 1\}$ such that*

$$\mathcal{L}_{\mathcal{D}}(h) = \Pr_{x \sim \mathcal{D}}\left[h(x) \neq c(x)\right] \leq 1/2 - \gamma.$$

A $\gamma$-weak learner thus achieves an advantage of $\gamma$ over random guessing when given $m_0$ samples. Note that $\mathcal{A}$ knows neither the distribution $\mathcal{D}$, nor the concrete concept $c \in C$ but achieves the advantage $\gamma$ for all $\mathcal{D}$ and $c$. We remark that in several textbooks (e.g. Mohri et al. [18]) a weak learner needs to work for any arbitrary $\delta > 0$ while Definition 1 only requires the existence of some $\delta_0$. Thus, every weak learner satisfying the definition of Mohri et al. also satisfies Definition 1, making our results more general.

In contrast to a weak learner, a strong learner can obtain arbitrarily high accuracy:

**Definition 2.** *Let $C \subseteq X \to \{-1, 1\}$ be a set of concepts and $\mathcal{A}$ a learning algorithm. We say that $\mathcal{A}$ is a strong learner for $C$, if for all $0 < \varepsilon, \delta < 1$, there is some number $m(\varepsilon, \delta)$ such that for every distribution $\mathcal{D}$ over $X$ and every concept $c \in C$, when given $m = m(\varepsilon, \delta)$ i.i.d. samples $S = x_1, \ldots, x_m$ from $\mathcal{D}$ together with their labels $c(x_1), \ldots, c(x_m)$, it holds with probability at least $1 - \delta$ over the sample $S$ and the randomness of $\mathcal{A}$, that $\mathcal{A}$ outputs a hypothesis $h : X \to \{-1, 1\}$ such that*

$$\mathcal{L}_{\mathcal{D}}(h) = \Pr_{x \sim \mathcal{D}}\left[h(x) \neq c(x)\right] \leq \varepsilon.$$

The definition of a strong learner is essentially identical to the classic notion of $(\varepsilon, \delta)$-PAC learning in the realizable setting. Unlike the $\gamma$-weak learner, we here require the learner to output a classifier with arbitrarily high accuracy ($\varepsilon$ small) and confidence ($\delta$ small) when given enough samples $S$.

Kearns and Valiant [15, 16] asked whether one can always obtain a strong learner when given access only to a $\gamma$-weak learner for a $\gamma > 0$. This was answered affirmatively by Schapire [21] and is the motivation behind the design of AdaBoost [7]. If we let $\mathcal{H}$ denote the set of hypotheses that a $\gamma$-weak learner may output from, then AdaBoost returns a *voting* classifier $f(x) = \text{sign}(\sum_{i=1}^{t} \alpha_i h_i(x))$ where each $h_i \in \mathcal{H}$ is the output of the $\gamma$-weak learner when trained on some carefully weighted version of the training set $S$ and each $\alpha_i$ is a real-valued weight. In terms of sample complexity $m(\varepsilon, \delta)$, the number of samples stated in Eq. (1) is sufficient for AdaBoost. There are several ways to prove this. For instance, it can be argued that when given $m$ samples, AdaBoost combines only $t = O(\gamma^{-2} \ln m)$ hypotheses $h_1, \ldots, h_t$ from $\mathcal{H}$ in order to produce an $f$ that perfectly classifies all the training data $S$, i.e. $f(x_i) = c(x_i)$ for all $x_i \in S$. Using that the class $\mathcal{H}^t$ can generate at most $O(\binom{m}{d}^t)$ distinct classifications of $m$ points (i.e. its growth function is bounded by this), one can intuitively invoke classic generalization bounds for PAC-learning in the realizable case to conclude that the hypothesis $f$ satisfies

$$\mathcal{L}_{\mathcal{D}}(f) \leq O\left(\frac{td\ln(m/d) + \ln(1/\delta)}{m}\right) = O\left(\frac{d\ln(m/d)\ln m}{\gamma^2 m} + \frac{\ln(1/\delta)}{m}\right) \tag{2}$$

with probability at least $1 - \delta$ over $S$ (and potentially the randomness of the weak learner). Using $\mathcal{L}_{\mathcal{D}}(f) = \varepsilon$ and solving Eq. (2) for $m$ gives the sample complexity stated in Eq. (1). This is the best sample complexity bound of any weak to strong learner prior to this work.

Our main upper bound result is a new algorithm with better sample complexity than AdaBoost and other weak to strong learners. It guarantees the following:

**Theorem 1.** *Assume we are given access to a $\gamma$-weak learner for some $0 < \gamma < 1/2$, using a base hypothesis set $\mathcal{H} \subseteq X \to \{-1, 1\}$ of VC-dimension $d$. Then there is a universal constant $\alpha > 0$ and an algorithm $\mathcal{A}$, such that $\mathcal{A}$ is a strong learner with sample complexity $m(\varepsilon, \delta)$ satisfying*

$$m(\varepsilon, \delta) \leq \alpha \cdot \left(\frac{d\gamma^{-2}}{\varepsilon} + \frac{\ln(1/\delta)}{\varepsilon}\right).$$

We remark that it is often required that a strong learner runs in polynomial time given a polynomial-time weak learner. This is indeed the case for our new algorithm.

Next, we complement our algorithm from Theorem 1 by the following lower bound:

**Theorem 2.** *There is a universal constant $\alpha > 0$ such that for all integers $d \in \mathbb{N}$ and every $2^{-d} < \gamma < 1/80$, there is a finite set $X$, a concept class $C \subset X \to \{-1, 1\}$ and a hypothesis set $\mathcal{H} \subseteq X \to \{-1, 1\}$ of VC-dimension at most $d$, such that for every $(\varepsilon, \delta)$ with $0 < \varepsilon < 1$ and $0 < \delta < 1/3$, there is a distribution $\mathcal{D}$ over $X$ such that the following holds:*

*1. For every $c \in C$ and every distribution $\mathcal{D}'$ over $X$, there is an $h \in \mathcal{H}$ with $\Pr_{x \sim \mathcal{D}'}[h(x) \neq c(x)] \leq 1/2 - \gamma$.*

*2. For any algorithm $\mathcal{A}$, there is a concept $c \in C$ such that $\mathcal{A}$ requires at least*

$$m \geq \alpha \cdot \left(\frac{d\gamma^{-2}}{\varepsilon} + \frac{\ln(1/\delta)}{\varepsilon}\right)$$

*samples $S$ and labels $c(S)$ to guarantee $\mathcal{L}_{\mathcal{D}}(h_S) \leq \varepsilon$ with probability at least $1 - \delta$ over $S$, where $h_S$ is the hypothesis produced by $\mathcal{A}$ on $S$ and $c(S)$.*

The first statement of Theorem 2 says that the concept class $C$ can be $\gamma$-weakly learned. The second point then states that any learner requires as many samples as our new algorithm. Moreover, the lower bound does not require the algorithm to even use a $\gamma$-weak learner, nor does it need to run in polynomial time for the lower bound to apply. Furthermore, the algorithm is even allowed to use the full knowledge of the set $C$ and the distribution $\mathcal{D}$. The only thing it does not know is which concept $c \in C$ provides the labels $c(S)$ to the training samples. The lower bound thus matches our upper bound except possibly for very small $\gamma < 2^{-d}$. We comment further on this case in Section 5.

In the next section, we present the overall ideas in our new algorithm, as well as a new generalization bound for voting classifiers that is key to our algorithm. Finally, we sketch the main ideas in the lower bound. Due to the space requirements, several proofs have been moved to the supplementary material.

## 1.2 Main ideas and voting classifiers

One of the key building blocks in our new algorithm is voting classifiers. To formally introduce voting classifiers, define from a hypothesis set $\mathcal{H} \subseteq \mathcal{X} \to \{-1, 1\}$ the set of all convex combinations $\Delta(\mathcal{H})$ of hypotheses in $\mathcal{H}$. That is, $\Delta(\mathcal{H})$ contains all functions $f$ of the form $f(x) = \sum_{i=1}^{t} \alpha_i h_i(x)$ with $\alpha_i > 0$ and $\sum_i \alpha_i = 1$. AdaBoost can be thought of as producing a voting classifier $g(x) = \text{sign}(f(x))$ for an $f \in \Delta(\mathcal{H})$ by appropriate normalization of the weights it uses.

Classic work on understanding the surprisingly high accuracy of AdaBoost introduced the notion of *margins* [2]. For a function $f \in \Delta(\mathcal{H})$, and a sample $x$ with label $y$, the margin of $f$ on $(x, y)$ is $yf(x)$. Notice that the margin is positive if and only if $\text{sign}(f(x))$ correctly predicts the label $y$ of $x$. It was empirically observed that AdaBoost produces voting classifiers $g(x) = \text{sign}(f(x))$ where $f$ has large margins. This inspired multiple generalization bounds based on the margins of a voting classifier, considering both the minimum and the $k$-th margin [12, 4, 3, 19, 20, 17]. The simplest bound when all margins are assumed to be at least $\gamma$, is Breiman's min margin bound:

**Theorem 3** (Breiman [4]). *Let $c \in \mathcal{X} \to \{-1, 1\}$ be an unknown concept, $\mathcal{H} \subseteq \mathcal{X} \to \{-1, 1\}$ a hypothesis set of VC-dimension $d$ and $\mathcal{D}$ an arbitrary distribution over $\mathcal{X}$. With probability at least $1 - \delta$ over a set of $m$ samples $S \sim \mathcal{D}^m$, it holds for every voting classifier $g(x) = \text{sign}(f(x))$ with $f \in \Delta(\mathcal{H})$ satisfying $c(x)f(x) \geq \gamma$ on all $x \in S$, that:*

$$\mathcal{L}_{\mathcal{D}}(g) = O\left(\frac{d \ln(m/d) \ln m}{\gamma^2 m}\right)$$

The resemblance to the generalization performance of AdaBoost in Eq. (2) is no coincidence. Indeed, a small twist to AdaBoost, presented in the AdaBoost$_\nu^*$ algorithm [20], ensures that the voting classifier produced by AdaBoost$_\nu^*$ from a $\gamma$-weak learner has all margins at least $\gamma/2$. This gives an alternative way of obtaining the previous best sample complexity in Eq. (1). We remark that more refined generalization bounds based on margins exist, such as the $k$-th margin bound by Gao and Zhou [8] which is known to be near-tight [10]. These bounds take the whole sequence of margins $c(x_i)f(x_i)$ of all samples $x_i \in S$ into account, not only the smallest. However, none of these bounds leads to better generalization from a $\gamma$-weak learner.

We note that the notion of margins has not only been considered in the context of boosting algorithms but also plays a key role in understanding the generalization performance of Support Vector Machines, see e.g. the recent works [14, 11] giving tight SVM generalization bounds in terms of margins.

In our new algorithm, we make use of a voting classifier with good margins as a subroutine. Concretely, we invoke AdaBoost$_\nu^*$ to obtain margins of at least $\gamma/2$ on all training samples. At first sight, this seems to incur logarithmic losses, at least if the analysis by Breiman is tight. Moreover, Grønlund et al. [9] proved a generalization lower bound showing that there are voting classifiers with margins $\gamma$ on all training samples, but where at least one of the logarithmic factors in the generalization bound must occur. To circumvent this, we first notice that the lower bound only applies when $m$ is sufficiently larger than $d\gamma^{-2}$. We carefully exploit this loophole and prove a new generalization bound for voting classifiers:

**Theorem 4.** *Let $c \in \mathcal{X} \to \{-1, 1\}$ be an unknown concept, $\mathcal{H} \subseteq \mathcal{X} \to \{-1, 1\}$ a hypothesis set of VC-dimension $d$ and $\mathcal{D}$ an arbitrary distribution over $\mathcal{X}$. There is a universal constant $\alpha > 0$, such that with probability at least $1 - \delta$ over a set of $m \geq \alpha(d\gamma^{-2} + \ln(1/\delta))$ samples $S \sim \mathcal{D}^m$, every voting classifier $g(x) = \text{sign}(f(x))$ with $f \in \Delta(\mathcal{H})$ satisfying $c(x)f(x) \geq \gamma$ on all $x \in S$ achieves*

$$\mathcal{L}_{\mathcal{D}}(g) \leq \tfrac{1}{200}.$$

The value $1/200$ is arbitrary and chosen to match the requirements in the proof of Theorem 1. Notice how our new generalization bound avoids the logarithmic factors when aiming merely at generalization error $1/200$. Breiman's bound would only guarantee that $d\gamma^{-2} \ln(1/\gamma) \ln(d/\gamma)$ samples suffice for such a generalization error. While the focus of previous work on generalization bounds was not on the constant error case, we remark that any obvious approaches to modify the previous proofs could perhaps remove the $\ln m$ factor but not the $\ln(m/d)$ factor. The $\ln(m/d)$ factor turns into $\Theta(\ln(1/\gamma))$ when solving for $m$ in $d \ln(m/d)/(\gamma^2 m) = 1/200$ and is insufficient for our purpose.

With the new generalization bound on hand, we can now construct our algorithm for producing a strong learner from a $\gamma$-weak learner. Here we use as template the sample optimal algorithm by

Hanneke [13] for PAC learning in the realizable case (which improved over a previous near-tight result by Simon [22]). Given a training set $S$, his algorithm carefully constructs a number of sub-samples $S_1, S_2, \ldots, S_k$ of $S$ and trains a hypothesis $h_i$ on each $S_i$ using empirical risk minimization. As the final classifier, he returns the voter $g(x) = \text{sign}\big((1/k) \sum_{i=1}^{k} h_i(x)\big)$.

For our new algorithm, we use Hanneke's approach to construct sub-samples $S_1, \ldots, S_k$ of a training set $S$. We then run AdaBoost$_\nu^*$ on each $S_i$ to produce a voting classifier $g_i(x) = \text{sign}(f_i(x))$ for an $f_i \in \Delta(\mathcal{H})$ with margins $\gamma/2$ on all samples in $S_i$. We finally return the voter $h(x) = \text{sign}((1/k) \sum_{i=1}^{k} g_i(x))$. Our algorithm thus returns a majority of majorities.

To prove that our algorithm achieves the desired sample complexity $m(\varepsilon, \delta)$ claimed in Theorem 1, we then revisit Hanneke's proof and show that it suffices for his argument that the base learning algorithm (in his case empirical risk minimization, in our case AdaBoost$_\nu^*$) achieves an error of at most $1/200$ when given $\tau$ samples. If this is the case, then his proof can be modified to show that the final error of the output voter drops to $O(\tau/m)$. Plugging in the $\tau = \alpha(d\gamma^{-2} + \ln(1/\delta))$ from our new generalization bound in Theorem 4 completes the proof.

Let us remark that a lower bound by Grønlund et al. [9] shows the existence of a voting classifier with simultaneously large margins and a generalization error with an additional log-factor. It is thus conceivable that a simple majority vote is not sufficient and a majority of majorities is indeed necessary, although the lower bound only guarantees the *existence* of a 'bad' voter with good margins and not that *all* such voters are 'bad'.

In the following, we start by proving our new generalization bound (Theorem 4) in Section 2. We then proceed in Section 3 to present our new algorithm and sketch the proof that it gives the guarantees in Theorem 1. Finally, in Section 4 we give the proof ideas of the lower bound in Theorem 2.

## 2 New margin-based generalization bounds for voting classifiers

In this section, we prove the new generalization bound stated in Theorem 4. For ease of notation, we write that $\mathcal{D}$ is a distribution over $\mathcal{X} \times \{-1, 1\}$ (and not just a distribution over $\mathcal{X}$) and implicitly assume that the label of each $x \in \mathcal{X}$ is $c(x)$ for the unknown concept $c$. Moreover, for a voting classifier $g(x) = \text{sign}(f(x))$ with $f \in \Delta(\mathcal{H})$, we simply refer to $f$ as the voting classifier and just remark that one needs to take the sign to make a prediction. Finally, we think of the sample $S$ as a set of pairs $(x_i, y_i)$ with $x_i \in \mathcal{X}$ and $y_i = c(x_i) \in \{-1, 1\}$.

The key step in the proof of Theorem 4 is to analyze the generalization performance for a voting classifier obtained by combining randomly drawn hypotheses among the hypotheses $h_1, \ldots, h_t$ making up a voting classifier $f = \sum_i \alpha_i h_i$ from $\Delta(\mathcal{H})$. We then relate that to the generalization performance of $f$ itself. Formally, we define a distribution $\mathcal{D}_{f,t}$ for every $f$ and look at a random hypothesis from $\mathcal{D}_{f,t}$. We start by defining this distribution.

Let $f(x) = \sum_h \alpha_h h(x) \in \Delta(\mathcal{H})$ be a voting classifier. Let $\mathcal{D}_f$ be the distribution over $\mathcal{H}$ (the base hypotheses used in $f$) where $h$ has probability $\alpha_h$. Consider drawing $t$ i.i.d. hypotheses $h'_1, \ldots, h'_t$ from $\mathcal{D}_f$ and then throwing away each $h'_i$ independently with probability $1/2$. Let $t'$ be the number of remaining hypotheses, denote them $h_1, \ldots, h_{t'}$, and let $g = \frac{1}{t'} \sum_{i=1}^{t'} h_i$. One can think of $g$ as a sub-sampled version of $f$ with replacement. Denote by $\mathcal{D}_{f,t}$ the distribution over $g$.

**Key properties of $\mathcal{D}_{f,t}$.** In the following, we analyze how a random $g$ from $\mathcal{D}_{f,t}$ behaves and show that while it behaves similar to $f$ it produces with good probability predictions that are big in absolute value (even if $f(x) \approx 0$). The proofs of the following three lemmas are given in the supplementary material. First, we note that predictions made by a random $g$ are often close to those made by $f$ and then observe that $g$ rarely makes predictions $g(x)$ that are small in absolute value.

**Lemma 1.** *For any $x \in \mathcal{X}$, any $f \in \Delta(\mathcal{H})$, and any $\mu > 0$:* $\Pr_{g \sim \mathcal{D}_{f,t}}[|f(x) - g(x)| \geq \mu] < 5e^{-\mu^2 t/32}$.

**Lemma 2.** *For any $x \in \mathcal{X}$, any $f \in \Delta(\mathcal{H})$, and any $\mu \geq 1/t$:* $\Pr_{g \sim \mathcal{D}_{f,t}}[|g(x)| \leq \mu] \leq 2\mu\sqrt{t}$.

Lemma 2 states that even if $f(x) \approx 0$ for an unseen sample $x$, $g(x)$ will still be large with good probability. Thus we can think of $g$ as having large margins (perhaps negative) also on unseen data. This is crucial for bounding the generalization error. The proof follows from an invocation of Erdős' improved Littlewood-Offord lemma [6].

As the last property, we look at the out-of-sample and in-sample error of a random $g$ and start with relating the generalization error of $f$ to that of a random $g$. To formalize this, define for any distribution $\mathcal{D}$, the loss $\mathcal{L}_{\mathcal{D}}^{t}(f) := \Pr_{(x,y)\sim\mathcal{D}, g\sim\mathcal{D}_{f,t}}\left[yg(x) \leq 0\right]$ and when writing $\mathcal{L}_{S}^{t}(f)$ we implicitly assume $S$ to also denote the uniform distribution over all $(x, y) \in S$. We then have:

**Lemma 3.** *For any distribution $\mathcal{D}$ over $\mathcal{X} \times \{-1, 1\}$, any $t \geq 36$ and any voting classifier $f \in \Delta(\mathcal{H})$ for a hypothesis set $\mathcal{H} \subset \mathcal{X} \rightarrow \{-1, 1\}$, we have $\mathcal{L}_{\mathcal{D}}(f) \leq 3\mathcal{L}_{\mathcal{D}}^{t}(f)$.*

Moreover, if $f$ has margins $\gamma$ on all training samples $(x, y) \in S$, then $g$ is correct on most of $S$ provided that we set $t$ big enough:

**Lemma 4.** *Let $S$ be a set of $m$ samples in $\mathcal{X} \times \{-1, 1\}$ and assume $f$ is a voting classifier with $yf(x) \geq \gamma$ for all $(x, y) \in S$. For $t \geq 1024\gamma^{-2}$, we have $\mathcal{L}_{S}^{t}(f) \leq 1/1200$.*

*Proof.* By Lemma 1, it holds for all $(x, y) \in S$, that $|f(x) - g(x)| \geq \gamma$ with probability at most $5\exp\left(-\gamma^{2}t/32\right) \leq 5e^{-32} \ll 1/1200$. Since $yf(x) \geq \gamma$, this implies $\mathrm{sign}\left(g(x)\right) = \mathrm{sign}\left(f(x)\right) = y$. $\square$

The last ingredient for the proof of Theorem 4 is to relate $\mathcal{L}_{S}^{t}(f)$ and $\mathcal{L}_{\mathcal{D}}^{t}(f)$. For the proof we use Lemma 2 to infer that with good probability $|g(x)| = \Omega(\gamma)$, i.e. has large absolute value. We use this to argue that $\mathrm{sign}(g)$ often belongs to a class with small VC-dimension and then apply a growth-function argument to relate $\mathcal{L}_{S}^{t}(f)$ and $\mathcal{L}_{\mathcal{D}}^{t}(f)$. Formally, we prove the following lemma.

**Lemma 5.** *Let $\mathcal{D}$ be an arbitrary distribution over $\mathcal{X} \times \{-1, 1\}$ and let $\mathcal{H} \subset \mathcal{X} \rightarrow \{-1, 1\}$ be a hypothesis set of VC-dimension $d$. There is a universal constant $\alpha > 0$ such that for any $t \in \mathbb{N}$ and any $m \geq \alpha td$, it holds that:*

$$\Pr_{S}\left[\sup_{f\in\Delta(\mathcal{H})} |\mathcal{L}_{S}^{t}(f) - \mathcal{L}_{\mathcal{D}}^{t}(f)| > \tfrac{1}{1200}\right] \leq \alpha \cdot \exp(-m/\alpha).$$

Before we prove Lemma 5, we show how to use it to prove Theorem 4. Since we are only aiming to prove the generalization of voting classifiers $f$ with $yf(x) \geq \gamma$ for all samples $(x, y) \in S$, Lemma 4 tells us that such $f$ have small $\mathcal{L}_{S}^{t}(f)$ when $t \geq 1024\gamma^{-2}$. We thus fix $t = 1024\gamma^{-2}$ and get that $\mathcal{L}_{S}^{t}(f) \leq 1/1200$ from Lemma 4. By Lemma 5, with probability at least $1 - \alpha\exp(-m/\alpha)$ over the sample $S$, we have for all $f \in \Delta(\mathcal{H})$ that $|\mathcal{L}_{S}^{t}(f) - \mathcal{L}_{\mathcal{D}}^{t}(f)| \leq 1/1200$ and thus $\mathcal{L}_{\mathcal{D}}^{t}(f) \leq 1/600$. Finally, Lemma 3 gives us that $\mathcal{L}_{\mathcal{D}}(f) \leq 3\mathcal{L}_{\mathcal{D}}^{t}(f)$ for all $f \in \Delta(\mathcal{H})$. Together we thus have $\mathcal{L}_{\mathcal{D}}^{t}(f) \leq 1/600 \Rightarrow \mathcal{L}_{\mathcal{D}}(f) \leq 1/200$ for any $m \geq \alpha td \geq \alpha' d\gamma^{-2}$ where $\alpha' > 0$ is a universal constant. By observing that $\alpha\exp(-m/\alpha) < \delta$ for $m \geq \alpha\ln(\alpha/\delta)$, this completes the proof of Theorem 4. What remains is thus to prove Lemma 5 which we do in the remainder of this section.

## 2.1  Relating $\mathcal{L}_{S}^{t}(f)$ and $\mathcal{L}_{\mathcal{D}}^{t}(f)$

The last remaining step to show Theorem 4 is thus to relate $\mathcal{L}_{S}^{t}(f)$ to $\mathcal{L}_{\mathcal{D}}^{t}(f)$, i.e. to prove Lemma 5. In the proof, we rely on the classic approach for showing generalization for classes $\mathcal{H}$ of bounded VC-dimension and introduce a *ghost* set that only exists for the sake of analysis. In addition to the sample $S$, we thus consider a *ghost* set $S'$ of another $m$ i.i.d. samples from $\mathcal{D}$. This allows us to prove:

**Lemma 6.** *For $m \geq 2400^{2}$ any $t$ and any $f$, it holds that:*

$$\Pr_{S}\left[\sup_{f\in\Delta(\mathcal{H})} |\mathcal{L}_{S}^{t}(f) - \mathcal{L}_{\mathcal{D}}^{t}(f)| > \tfrac{1}{1200}\right] \leq 2 \cdot \Pr_{S,S'}\left[\sup_{f\in\Delta(\mathcal{H})} |\mathcal{L}_{S}^{t}(f) - \mathcal{L}_{S'}^{t}(f)| > \tfrac{1}{2400}\right].$$

As the proof is standard, it can be found in the supplementary material.

We thus only need to bound $\Pr_{S,S'}[\sup_{f\in\Delta(\mathcal{H})} |\mathcal{L}_{S}^{t}(f) - \mathcal{L}_{S'}^{t}(f)| > 1/2400]$. To do this, consider drawing a data set $P$ of $2m$ i.i.d. samples from $\mathcal{D}$ and then drawing $S$ as a set of $m$ uniform samples from $P$ without replacement and letting $S'$ be the remaining samples. Then $S$ and $S'$ have the same distribution as if they were drawn as two independent sets of $m$ i.i.d. samples each. From here on, we thus think of $S$ and $S'$ as being sampled via $P$.

Now consider a fixed set $P$ in the support of $\mathcal{D}^{2m}$ and define $\Delta_{\delta}^{\mu}(\mathcal{H}, P)$ as the set of voting classifiers $f \in \Delta(\mathcal{H})$ for which $\Pr_{(x,y)\sim P}[|f(x)| \geq \mu] \geq 1 - \delta$. These are the voting classifiers that make

predictions of large absolute value on most of *both* S and S' (if $\delta \ll 1/2$). The crucial point, and the whole reason for introducing $g$, is that regardless of what $f$ is, a random $g \sim \mathcal{D}_{f,t}$ often lies in the set $\Delta_\delta^\mu(\mathcal{H}, P)$:

**Lemma 7.** *For any data set P, parameters $0 < \delta < 1$ and t, and every $\mu \le \delta/(9600\sqrt{t})$, we have* $\Pr_{g \sim \mathcal{D}_{f,t}}[g \notin \Delta_\delta^\mu(\mathcal{H}, P)] \le 1/4800$.

*Proof.* Define an indicator $X_i$ for each $(x_i, y_i) \in P$ taking the value 1 if $|g(x_i)| \le \mu$. By Lemma 2, we have $\mathbb{E}[\sum_i X_i] \le |P| 2\mu\sqrt{t} \le |P| \delta/4800$. By Markov's inequality $\Pr[\sum_i X_i \ge \delta|P|] \le 1/4800$. □

If we had just considered $f$, we had no way of arguing that $f$ makes predictions of large absolute value on S', since the only promise we are given is that it does so on S. That $g$ makes predictions of large absolute value even outside of S is crucial for bounding the generalization error in the following.

Let us now define $\hat{\Delta}_\delta^\mu(P) = \text{sign}\left(\Delta_\delta^\mu(\mathcal{H}, P)\right)$ which means that $\hat{\Delta}_\delta^\mu(P)$ contains all the hypotheses that are obtained by voting classifiers in $\Delta_\delta^\mu(\mathcal{H}, P)$ when taking the sign. Since $g$ is in $\Delta_\delta^\mu(\mathcal{H}, P)$ except with probability $1/4800$ by Lemma 7, we can prove:

**Lemma 8.** *For any $0 < \delta < 1$, every t, and every $\mu \le \delta/(9600\sqrt{t})$, we have*

$$\Pr_{S,S'}\left[\sup_{f \in \Delta(\mathcal{H})} \left|\mathcal{L}_S^t(f) - \mathcal{L}_{S'}^t(f)\right| > \tfrac{1}{2400}\right] \le \sup_P 2|\hat{\Delta}_\delta^\mu(P)| \exp\left(-2m/9600^2\right).$$

Again, the proof of the lemma can be found in the supplementary material. What Lemma 8 gives us, is that it relates the generalization error to the growth function $|\hat{\Delta}_\delta^\mu(P)|$. The key point is that $\hat{\Delta}_\delta^\mu(P)$ was obtained from voting classifiers with predictions of large absolute value on all but a $\delta$ fraction of points in P. This implies that we can bound the VC-dimension of $\hat{\Delta}_\delta^\mu(P)$ when restricted to the point set P using Rademacher complexity:

**Lemma 9.** *Let $\mathcal{H}$ be a hypothesis set of VC-dimension d. For any $\delta, \mu > 0$ and point set P, we have that the largest subset P' of P that $\hat{\Delta}_\delta^\mu(P) = \text{sign}\left(\Delta_\delta^\mu(\mathcal{H}, P)\right)$ can shatter, has size at most $|P'| = d' < \max\{2\delta|P|, 4\alpha^2\mu^{-2}d\}$, where $\alpha > 0$ is a universal constant.*

*Proof.* Recall that the VC-dimension of $\mathcal{H}$ is d. Thus the Rademacher complexity of $\mathcal{H}$ for any point set P' is:

$$\mathbb{E}_{\sigma \in P' \to \{-1,1\}}\left[\frac{1}{|P'|} \sup_{h \in \mathcal{H}}\left|\sum_{x \in P'} h(x)\sigma(x)\right|\right] < \alpha\sqrt{\frac{d}{|P'|}}$$

for a universal constant $\alpha > 0$ (see e.g. [23]). Assume $P' \subseteq P$ with $|P'| = d'$ can be shattered. Fix any labeling $\sigma \in P' \to \{-1, 1\}$. Let $h_\sigma \in \hat{\Delta}_\delta^\mu(P)$ be the hypothesis generating the dichotomy $\sigma$ (which exists since P' is shattered). Since $h_\sigma \in \hat{\Delta}_\delta^\mu(P)$, there must be some $g \in \Delta_\delta^\mu(\mathcal{H}, P)$ such that $h_\sigma = \text{sign}(g)$ on the point set P'. If $|P'| \ge 2\delta|P|$, then by definition of $\Delta_\delta^\mu(\mathcal{H}, P)$, there are at least $|P'| - \delta|P| \ge |P'|/2$ points $x \in P'$ for which $|g(x)| \ge \mu$. This means that $(1/|P'|)\sum_{x \in P'} g(x)\sigma(x) \ge (1/2)\mu$. But $g(x)$ is a convex combination of hypotheses from $\mathcal{H}$, hence there is also a hypothesis $h \in \mathcal{H}$ for which $(1/|P'|)\sum_{x \in P'} h(x)\sigma(x) \ge (1/2)\mu$. Since this holds for all $\sigma$, by the bound on the Rademacher complexity, we conclude $\alpha\sqrt{d/|P'|} > (1/2)\mu \implies |P'| < 4\alpha^2\mu^{-2}d$. We thus conclude that the largest set that $\hat{\Delta}_\delta^\mu(P)$ can shatter, has size less than $\max\{2\delta|P|, 4\alpha^2\mu^{-2}d\}$. □

We remark that it was crucial to introduce the random hypothesis $g$, since all we are promised about the original hypothesis $f$ is that it has large margins on S, i.e. on only half the points in P. That case would correspond to $\delta = 1/2$ in Lemma 9 and would mean that we could potentially shatter all of P. In order for the bound to be useful, we thus need $\delta \ll 1/2$ and thus large margins on much more than half of P (which we get by using $g$).

For a $0 < \delta < 1$ to be determined, let us now fix $\mu = \delta/(9600\sqrt{t})$ and assume that the number of samples $m$ satisfies $m \ge \max\{\alpha^2\mu^{-2}d/\delta, 2400^2\}$ where $\alpha$ is the constant from Lemma 9. By Lemma 8, we have

$$\Pr_{S,S'}\left[\sup_{f \in \Delta(\mathcal{H})} |\mathcal{L}_S^t(f) - \mathcal{L}_{S'}^t(f)| > \tfrac{1}{2400}\right] \le \sup_P 2|\hat{\Delta}_\delta^\mu(P)| \exp(-2m/9600^2).$$

---

**Algorithm 1:** *Sub-Sample*$(A, B)$  (Hanneke [13])

---

**Input:** Sets $A$ and $B$

1 **if** $|A| \leq 3$ **then**                                     `// stop when A is too small to recurse`
2    |   **return** $A \cup B$
3 **else**
4    |   Let $A_0$ denote the first $|A| - 3\lfloor |A|/4 \rfloor$ elements of $A$,                    `// split A evenly`
            $A_1$ the next $\lfloor |A|/4 \rfloor$ elements,
            $A_2$ the next $\lfloor |A|/4 \rfloor$ elements, and
            $A_3$ the remaining $\lfloor |A|/4 \rfloor$ elements.
5    |   **return** *Sub-Sample*$(A_0, A_2 \cup A_3 \cup B) \cup$                 `// recurse in leave-one-out fashion`
           *Sub-Sample*$(A_0, A_1 \cup A_3 \cup B) \cup$
           *Sub-Sample*$(A_0, A_1 \cup A_2 \cup B)$

---

---

**Algorithm 2:** Optimal weak-to-strong learner

---

**Input:** Set $S$ of $m$ samples.

1 $\{C_1, \ldots, C_k\} =$ *Sub-Sample*$(S, \emptyset)$                `// create highly overlapping subsamples of S`
2 **for** $i = 1, \ldots, k$ **do**
3    |   $h_i = \mathcal{A}_\gamma^*(C_i)$                        `// run AdaBoost*ᵥ on all those sub-samples`
4 **return** $h(x) = \text{sign}\left( \sum_{i=1}^k h_i(x) \right)$.                `// return unweighted majority vote`

---

Lemma 9 gives us that the largest subset $P' \subseteq P$ that $\hat{\Delta}_\delta^\mu(P)$ shatters has size at most $d' < \max\{2\delta|P|, 4\alpha^2\mu^{-2}d\}$. By our assumption on $m$, the term $2\delta|P| = 4\delta m$ is at least $4c^2\mu^{-2}d$ and thus $2\delta|P| = 4\delta m$ takes the maximum in the bound on $d'$. By the Sauer-Shelah lemma, we have that $|\hat{\Delta}_\delta^\mu(P)| \leq \sum_{i=0}^{4\delta m-1} \binom{2m}{i}$. For $\delta \leq 1/4$, this is at most $\binom{2m}{4\delta m} \leq (e\delta^{-1}/2)^{4\delta m} = \exp\left(4\delta m \ln(e\delta^{-1}/2)\right)$.

As conclusion we have:

$$\Pr_{S,S'}\left[ \sup_{f \in \Delta(\mathcal{H})} |\mathcal{L}_S^t(f) - \mathcal{L}_{S'}^t(f)| > \tfrac{1}{2400} \right] \leq 2\exp(4\delta m \ln(e\delta^{-1}/2)) \exp(-2m/9600^2).$$

Let us now fix $\delta = 10^{-10}$. We then have

$$\begin{aligned}
&2\exp\left(4\delta m \ln(e\delta^{-1}/2)\right) \exp(-2m/9600^2) \\
=\ &2\exp\left(m(4\delta \ln(e\delta^{-1}/2) - 2/9600^2)\right) \\
\leq\ &2\exp\left(-m/10^8\right)
\end{aligned}$$

where the last step is a numerical calculation. By Lemma 6, this in turn implies:

$$\Pr_S\left[ \sup_{f \in \Delta(\mathcal{H})} \left|\mathcal{L}_S^t(f) - \mathcal{L}_\mathcal{D}^t(f)\right| > \tfrac{1}{1200} \right] \leq 4\exp(-m/10^8).$$

Since we only required $m \geq \max\{\alpha^2\mu^{-2}d/\delta, 2400^2\}$ and we had $\mu = \delta/(9600\sqrt{t})$, this is satisfied for $m \geq \alpha' t d$ for a large enough constant $\alpha' > 0$. This completes the proof of Lemma 5 and thus also finishes the proof of Theorem 4.

## 3 Weak to strong learning

In this section, we give our algorithm for obtaining a strong learner from a $\gamma$-weak learner with optimal sample complexity and sketch the main proof idea.

The algorithm obtaining the guarantees of Theorem 1 is as follows: Let $\mathcal{A}_\gamma^*$ be an algorithm that on a sample $S$ outputs a classifier $g = \text{sign}(f)$, where $f$ is a voting classifier with margins at least $\gamma/2$ on all samples in $S$ such as AdaBoost$_\gamma^*$ [20]. Given a set $S$ of $m$ i.i.d. samples from an unknown distribution $\mathcal{D}$, we run $\mathcal{A}_\gamma^*$ on a number of samples $C_1, C_2, \ldots, C_k \subset S$ obtaining hypotheses $h_1, h_2, \ldots, h_k$. We then return the (unweighted) majority vote among $h_1, \ldots, h_k$ as our final hypothesis $h^*$. The subsets $C_i$ are chosen by the algorithm *Sub-Sample* (shown in Algorithm 1) as in the optimal PAC learning algorithm by Hanneke [13]. The final algorithm (Algorithm 2) calls $\mathcal{A}_\gamma^*$ on all subsets

returned by Algorithm 1 and returns the majority vote. Note that the final hypothesis returned by Algorithm 2 is a majority of majorities since $\mathcal{A}_\gamma^*$ already returns a voting classifier. Let us also remark that in Theorem 4 we have a failure probability $\delta_0 > 0$, while the analysis of AdaBoost$_\gamma^*$ assumes $\delta_0 = 0$, i.e. that the weak learner always achieves an advantage of at least $\gamma$. If one knows $\gamma$ in advance, this is not an issue as AdaBoost$_\gamma^*$ only calls the weak learner on distributions over the training data $S$ and one can thus compute the advantage from the training data. After in expectation $1/(1 - \delta_0)$ invocations of the weak learner, we thus get a hypothesis with advantage $\gamma$.

In total there are $k = 3^{\lceil \log_4(m) \rceil} \approx m^{0.79}$ calls to the weak learner, each with a sub-sample of linear size. Since AdaBoost$_\gamma^*$ runs in polynomial time on its input, given that the weak learner is polynomial, Algorithm 2 is polynomial under the same condition.

The formal proof that Algorithm 2 has the guarantees of Theorem 1 is given in the supplementary material due to space constraints. It follows the proof of Hanneke [13] pretty much uneventfully, although carefully using that a generalization error of $1/200$ suffices for each call of $\mathcal{A}_\gamma^*$. The key observation is that each of the recursively generated sub-samples in Algorithm 1 leaves out a subset $A_i$ of the training data, whereas the two other recursive calls always include all of $A_i$ in their sub-samples. If one considers a hypothesis $h$ trained on the data leaving out $A_i$, then $A_i$ serves as an independent sample from $\mathcal{D}$. This implies that if $h$ has large error probability over $\mathcal{D}$, then many of the samples in $A_i$ will be classified incorrectly by $h$. Now, since the two other recursive calls always include $A_i$, any hypothesis $h'$ trained on a sub-sample from those calls will have margin at least $\gamma/2$ on all points misclassified by $h$ in $A_i$. But the generalization bound in Theorem 2 then implies that $h'$ makes a mistake only with probability $1/200$ on *the conditional distribution* $\mathcal{D}(\,\cdot\mid h \text{ errs})$. Thus, the probability that they both err at the same time is at most the probability that $h$ errs, times $1/200$. Applying this reasoning inductively gives the conclusion that it is very unlikely that the majority of all trained hypotheses err at the same time which then finishes the proof.

## 4 Lower bound

In this section, we sketch the proof of Theorem 2 which gives a lower bound that matches the sample complexity from Theorem 1. The full proof is given in the supplementary material. Informally, Theorem 2 says that there exists a weakly learnable concept class $C$ such that the hypothesis $\mathcal{A}(S)$ of any learning algorithm $\mathcal{A}$ satisfies

$$\mathcal{L}_\mathcal{D}(\mathcal{A}(S)) \geq \alpha \left( \frac{d}{\gamma^2 m} + \frac{\ln(1/\delta)}{m} \right).$$

Here, the term $\ln(1/\delta)/m$ follows from previous work. In particular, we could let $C = \mathcal{H}$ and invoke the tight lower bounds for PAC-learning in the realizable setting [5].

Thus, we let $\delta = 1/3$ and only prove that the loss of $\mathcal{A}(S)$ is at least $\alpha d/(\gamma^2 m)$ with probability $1/3$ over $S$ when $|S| = m$ for some weakly learnable concept class $C$. This proof uses a construction from Grønlund et al. [9] to obtain a hypothesis set $\mathcal{H}$ over a domain $\mathcal{X} = \{x_1, \ldots, x_u\}$ of cardinality $u = \alpha d \gamma^{-2}$ such that a constant fraction of all concepts in $\mathcal{X} \to \{-1, 1\}$ can be $\gamma$-weakly learned from $\mathcal{H}$. We then create a distribution $\mathcal{D}$ where the first point $x_1$ is sampled with probability $1 - u/(4m)$ and with the remaining probability, we receive a uniform sample among $x_2, \ldots, x_u$. The key point is that we only expect to see $1 + m \cdot u/(4m) \approx u/4$ distinct points from $\mathcal{X}$ in a sample $S$ of cardinality $m$. Thus, if we consider a random concept that can be $\gamma$-weakly learned, the labels it assigns to points not in the sample are almost uniform random and independent. This in turn implies that the best any algorithm $\mathcal{A}$ can do is to guess the labels of points in $\mathcal{X} \setminus S$. In that way, $\mathcal{A}$ fails with constant probability if we condition on receiving a sample other than $x_1$. This happens with probability $u/(4m) = 4\alpha d \gamma^{-2}/m$ and the lower bound follows.

To formally carry out the intuitive argument above, we first argue that for a random concept $c \in C$, the Shannon entropy of $c$ is high, even conditioned on $S$ and the labels $c(S)$. Secondly, we argue that if $\mathcal{A}(S)$ has a small error probability under $\mathcal{D}$, then it must be the case that the hypothesis $\mathcal{A}(S)$ reveals a lot of information about $c$, i.e. the entropy of $c$ is small conditioned on $\mathcal{A}(S)$. Since $\mathcal{A}(S)$ is a function of $S$ and $c(S)$, the same holds if we condition on $S$ and $c(S)$. This contradicts that $c$ has high entropy and thus we conclude that $\mathcal{A}(S)$ cannot have a small error probability.

# 5 Conclusion

Overall, we presented a new weak to strong learner with a sample complexity that removes two logarithmic factors from the best-known bound. By accompanying the algorithm with a matching lower bound for all $d$ and $2^{-d} < \gamma < 1/80$, we showed that the achieved sample complexity of our algorithm is indeed optimal. Our algorithm uses the same sub-sampling technique as Hanneke [13] and computes a voting classifier with large margins for each sample for example with AdaBoost$_\nu^*$ [20]. The analysis of our algorithm uses a new generalization bound for voting classifiers with large margins.

Although we determined the exact sample complexity of weak to strong learning (up to multiplicative constants), there are a few connected open problems. Currently, our construction uses $3^{\log_4(m)} \approx m^{0.79}$ many sub-samples of linear size as input to AdaBoost$_\nu^*$. For very large datasets, it would be great to reduce the number and size of these calls. We conjecture that the most promising way to do so is to revisit Hanneke's optimal PAC learner and improve the sub-sampling strategy there. This could lead to an improvement for the realizable case as well as to faster weak-to-strong learners.

Next, the output of our algorithm is a majority vote over majority voters. It is unclear whether a simple voter could achieve the same bounds. We believe that a majority of majorities is actually necessary. This is supported by a lower bound showing that there are voters with large margin and poor generalization (paying a logarithmic factor) and thus the learning algorithm has to avoid this 'bad' voter. We currently see no indication of how a variant of AdaBoost could do that.

For the regime of $\gamma < 2^{-d}$ which our lower bound does not capture, is it possible to use fewer samples? A recent result by Alon et al. [1] might suggest so. Concretely, they show that if a concept class $C$ can be $\gamma$-weak learned from a base hypothesis set $\mathcal{H}$ of VC-dimension $d$, then the VC-dimension of $C$ is no more than $O_d(\gamma^{-2+2/(d+1)})$, where $O_d(\cdot)$ hides factors only depending on $d$. Interestingly, the part $\gamma^{2/(d+1)}$ becomes non-trivial precisely when our lower bound stops applying, i.e. when $\gamma < 2^{-d}$. This could hint at a possibly better dependency on $\gamma$ for $\gamma < 2^{-d}$.

We have a new generalization bound for large-margin classifiers, which is better than the $k$-th margin bound (Gao and Zhou [8]) for constant error. Can the $k$-th margin bound in general be improved, perhaps by one logarithmic factor? One of our key new ideas is the application of the Littlewood-Offord lemma which might also be helpful for the more general case of non-constant error.

## Acknowledgments and Disclosure of Funding

This work was supported by the Digital Research Centre Denmark (DIREC) and by the Independent Research Fund Denmark (DFF) Sapere Aude Research Leader grant No 9064-00068B.

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
