# Appendix of the paper: Optimal Weak to Strong Learning

## A    Weak to strong learning

The following theorem is essentially a restatement of Theorem 1 from the main paper.

**Theorem A.1.** *Assume we are given access to a $\gamma$-weak learner for a $0 < \gamma < 1/2$, using base hypothesis set $\mathcal{H} \subseteq X \rightarrow \{-1, 1\}$ of VC-dimension $d$. Then there is a universal constant $\alpha > 0$ and an algorithm $\mathcal{A}$, such that for every $0 < \delta < 1$ and every distribution $\mathcal{D}$ over $X \times \{-1, 1\}$, it holds with probability at least $1 - \delta$ over a set of $m$ samples $S \sim \mathcal{D}^m$, that $\mathcal{A}$ on $S$ outputs a classifier $h_S = \mathcal{A}(S) \in X \rightarrow \{-1, 1\}$ with*

$$\mathcal{L}_\mathcal{D}(h_S) \leq \alpha \cdot \frac{d\gamma^{-2} + \ln(1/\delta)}{m}.$$

Theorem 1 of the main paper follows by setting $\varepsilon = \mathcal{L}_\mathcal{D}(h_S)$ and solving for $m$ and letting the label in the distribution $\mathcal{D}$ be $c(x)$ for every $x \in X$. The algorithm that obtains the guarantees has been described in the main paper. We thus only present (again) the two algorithms (Algorithm 1 and Algorithm 2), as well as AdaBoost$_\nu^*$ (Algorithm A.1) by Rätsch et al. [20] that achieves almost optimal margins and is used in Algorithm 2.

In the remainder of the section, we prove that Algorithm 2 has the guarantees of Theorem A.1. The proof follows that of Hanneke [13] pretty much uneventfully, although carefully using that a generalization error of $1/200$ suffices. For simplicity, we assume $m$ is a power of 4. This can easily be ensured by rounding $m$ down to the nearest power of 4 and ignoring all excess samples. This only affects the generalization bound by a constant factor. With $m$ being a power of 4 we can observe from Algorithm 1 that the cardinalities of all recursively generated sets $A_0$ (which are the input to the next level of the recursion) are also powers of 4. Hence we can ignore all roundings.

### A.1    Proof of Optimal Strong Learning

Let $C \subseteq X \rightarrow \{-1, 1\}$ be a concept class and assume there is a $\gamma$-weak learner for $C$ using hypothesis set $\mathcal{H}$ of VC-dimension $d$. Let $\mathcal{A}_\nu^*$ be an algorithm that on a sample $S$ consistent with a concept $c \in C$, computes a voting classifier $f \in \Delta(\mathcal{H})$ with $yf(x) \geq \gamma/2$ for all $(x, y) \in S$ and returns as its output hypothesis $g(x) = \text{sign}(f(x))$. We could e.g. let $\mathcal{A}_\nu^*$ be AdaBoost$_\nu^*$. For a sample $S$, we use the notation $\mathcal{M}_\gamma(S)$ to denote the set of hypotheses $g(x) = \text{sign}(f(x))$ for an $f \in \Delta(\mathcal{H})$ satisfying $yf(x) \geq \gamma$ for all $(x, y) \in S$. The set $\mathcal{M}_\gamma(S)$ is thus the set of all voting classifiers obtained by taking the sign of a voter that has margins at least $\gamma$ on all samples in $S$. By definition, the output hypothesis $g$ of $\mathcal{A}_\nu^*$ on a set of samples $S$ always lies in $\mathcal{M}_{\gamma/2}(S)$.

Let $c \in C$ be an unknown concept in $C$ and let $\mathcal{D}$ be an arbitrary distribution over $X$. Let $S = \{(x_i, c(x_i))\}_{i=1}^m \in (X \times \{-1, 1\})^m$ be a set of $m$ samples with each $x_i$ an i.i.d. sample from $\mathcal{D}$. Let $S_{1:k}$ denote the first $k$ samples of $S$. Let $c' \geq 4$ be a constant to be determined later. We will prove by induction that for every $m' \in \mathbb{N}$ that is a power of 4, for every $\delta' \in (0, 1)$, and every finite sequence $B'$ of samples in $X \times \{-1, 1\}$ with $y_i = c(x_i)$ for each $(x_i, y_i) \in B'$, with probability at least $1 - \delta'$, the classifier

$$\hat{h}_{m',B'} = \text{sign}\left(\sum_{C_i \in \text{Sub-Sample}(S_{1:m'}, B')} \mathcal{A}_\nu^*(C_i)\right)$$

satisfies

$$\mathcal{L}_\mathcal{D}(\hat{h}_{m',B'}) \leq \frac{c'}{m'}\left(d\gamma^{-2} + \ln(1/\delta')\right). \tag{3}$$

The conclusion of Theorem A.1 follows by letting $B' = \emptyset$ and $m' = m$ (and recalling that we assume $m$ is a power of 4). Thus what remains is to give the inductive proof.

As the base case, consider any $m' \in \mathbb{N}$ with $m' \leq c'$ and $m'$ a power of 4. In this case, the bound $c'(d\gamma^{-2} + \ln(1/\delta'))/m'$ is at least $d\gamma^{-2} \geq 1$ and $\mathcal{L}_\mathcal{D}(\hat{h}_{m',B'}) \leq 1$ obviously holds.

For the inductive step, take as inductive hypothesis that, for some $m \in \mathbb{N}$ with $m > c'$ and $m$ a power of 4, it holds for all $m' \in \mathbb{N}$ with $m' < m$ and $m'$ a power of 4, that for every $\delta' \in (0, 1)$ and every

**Algorithm A.1:** AdaBoost$_\nu^*$ [20]

**Input:** training set $S = \{(x_1, y_1), \ldots, (x_m, y_m)\}$
    number of rounds $T$
    desired accuracy $\nu$
**Result:** An ensemble hypothesis $H_{\text{out}}$ with almost optimal margins

1   $\mathcal{D}^{(1)} \leftarrow \left(\frac{1}{m}, \ldots \frac{1}{m}\right)$            `// uniform initialization of` $\mathcal{D}$

2 **for** $t \in \{1, \ldots, T\}$ **do**

3     $h_t \leftarrow \text{WL}(\mathcal{D}^{(t)}, S)$            `// invoke weak learner`

4     $\gamma_t \leftarrow \sum_{i=1}^m \mathcal{D}_i^{(t)} y_i h_t(x_i)$            `// average margin of` $h_t$

5     **if** $|\gamma_t| = 1$ **then**

         `/*` $h_t$ `is consistent` $\Rightarrow$ `taking only` $h_t$ `as 'ensemble' maximizes the margin`    `*/`

6         $w_1 \leftarrow \text{sign}(\gamma_t), \quad h_1 \leftarrow h_t, \quad T \leftarrow 1$

7         **break**

8     $\gamma_t^{\min} \leftarrow \min_{r \in [t]} \gamma_r,$            `// update assumed advantage`

9     $\rho_t^{\min} \leftarrow \gamma_t^{\min} - \nu$

10    $w_t = \frac{1}{2} \ln \frac{1+\gamma_t}{1-\gamma_t} - \frac{1}{2} \ln \frac{1+\rho_t}{1-\rho_t}$            `// weight for the current hypothesis`

11    **for** $i \in \{1, \ldots, m\}$ **do**

12      $\mathcal{D}_i^{(t+1)} \leftarrow \dfrac{\mathcal{D}_i^{(t)} \exp\left(-w_t y_i h_t(x_i)\right)}{\sum_{j=1}^m \mathcal{D}_i^{(t)} \exp\left(-w_t y_j h_t(x_j)\right)}$            `// update` $\mathcal{D}$

13 **return** $f_{out}(x) = \frac{1}{\sum_{i=1}^T |w_i|} \sum_{t=1}^T w_t h_t(x)$         `// (normalized) weighted majority vote`

finite sequence $B'$ of samples in $\mathcal{X} \times \{-1, 1\}$ with $y_i = c(x_i)$ for each $(x_i, y_i) \in B'$, with probability at least $1 - \delta'$, Eq. (3) holds. We need to prove that the inductive hypothesis also holds for $m' = m$.

Fix a $\delta \in (0, 1)$ and any finite sequence $B$ of points in $\mathcal{X} \times \{-1, 1\}$ with $y_i = c(x_i)$ for each $(x_i, y_i)$ in $B$. Since $m > c' \geq 4$, we have that *Sub-Sample*$(S_{1:m}, B)$ returns in Step 5 of Algorithm 1. Let $A_0, A_1, A_2, A_3$ be as defined in Step 4 of Algorithm 1. Also define $B_1 = A_2 \cup A_3 \cup B$, $B_2 = A_1 \cup A_3 \cup B$, $B_3 = A_1 \cup A_2 \cup B$, and for each $i \in \{1, 2, 3\}$, denote

$$h_i = \text{sign}\left(\sum_{C_i \in \textit{Sub-Sample}(A_0, B_i)} \mathcal{A}_\nu^*(C_i)\right).$$

Note that the $h_i$'s correspond to the majority vote classifiers trained on the sub-samples of the three recursive calls in Algorithm 1. Moreover, notice that $h_i = \hat{h}_{m/4, B_i}$. Therefore, the inductive hypothesis may be used on $h_1, h_2, h_3$ to conclude that for each $i \in \{1, 2, 3\}$, there is an event $E_i$ of probability at least $1 - \delta/9$, on which

$$\mathcal{L}_\mathcal{D}(h_i) \leq \frac{c'}{|A_0|}\left(d\gamma^{-2} + \ln(9/\delta)\right) \leq \frac{4c'}{m}\left(d\gamma^{-2} + \ln(1/\delta) + 3\right) \leq \frac{12c'}{m}\left(d\gamma^{-2} + \ln(1/\delta)\right). \quad (4)$$

Here we chose the probability $1 - \delta/9$ in order to perform a union bound in the end of the induction step which is possible since the inductive hypothesis holds for every $\delta'$. Next, define $\text{Err}(h_i)$ as the set of points $x \in \mathcal{X}$ for which $h_i(x) \neq c(x)$. Now fix an $i \in \{1, 2, 3\}$ and denote by $\{(Z_{i,1}, c(Z_{i,1})), \ldots, (Z_{i,N_i}, c(Z_{i,N_i}))\} = A_i \cap (\text{Err}(h_i) \times \{-1, 1\})$, where $N_i = |A_i \cap (\text{Err}(h_i) \times \{-1, 1\})|$. Said in words, the set $\{(Z_{i,j}, c(Z_{i,j}))\}_{j=1}^{N_i}$ is the subset of samples in $A_i$ on which $h_i$ makes a mistake. Notice that $h_i$ is not trained on any samples from $A_i$ ($B_i$ excludes $A_i$), hence $h_i$ and $A_i$ are independent. Therefore, given $h_i$ and $N_i$, the samples $Z_{i,1}, \ldots, Z_{i,N_i}$ are conditionally independent samples with distribution $\mathcal{D}(\cdot \mid \text{Err}(h_i))$ (provided $N_i > 0$). From Theorem 6 in the main paper, we get that there is an event $E_i'$ of probability at least $1 - \delta/9$, such that if $N_i \geq c''(d\gamma^{-2} + \ln(1/\delta))$, then every $h \in \mathcal{M}_{\gamma/2}\left(\{(Z_{i,j}, c(Z_{i,j}))\}_{j=1}^{N_i}\right)$ satisfies

$$\mathcal{L}_{\mathcal{D}(\cdot \mid \text{Err}(h_i))}(h) \leq \frac{1}{200}.$$

Note that this is a key step where our proof differs from Hanneke's original proof since we exploit that a bound of $\frac{1}{200}$ on the generalization error suffices for the rest of the proof. We continue by observing

that for each $j \in \{1, 2, 3\} \setminus \{i\}$, the set $B_j$ contains $A_i$ and this remains the case in all recursive calls of *Sub-Sample*$(A_0, B_i)$. Thus for $\{C_1, \ldots, C_k\} = $ *Sub-Sample*$(A_0, B_j)$, it holds for all $C_k$ that $\mathcal{A}_\nu^*(C_k) \in \mathcal{M}_{\gamma/2}(B_j) \Rightarrow \mathcal{A}_\nu^*(C_k) \in \mathcal{M}_{\gamma/2}(A_0) \Rightarrow \mathcal{A}_\nu^*(C_k) \in \mathcal{M}_{\gamma/2}(\{(Z_{i,j}, c(Z_{i,j}))\}_{j=1}^{N_i})$. Thus on the event $E_i'$ defined above, if $N_i > c''(d\gamma^{-2} + \ln(1/\delta))$, then it holds for all $j \in \{1, 2, 3\} \setminus \{i\}$ and all $C_k \in $ *Sub-Sample*$(A_0, B_j)$, that the hypothesis $h = \mathcal{A}_\nu^*(C_k)$ satisfies

$$\Pr_{x \sim \mathcal{D}} \left[ h_i(x) \neq c(x) \wedge h(x) \neq c(x) \right] = \mathcal{L}_{\mathcal{D}}(h_i) \cdot \mathcal{L}_{\mathcal{D}(\cdot|\mathrm{Err}(h_i))}(h)$$

$$\leq \tfrac{1}{200} \mathcal{L}_{\mathcal{D}}(h_i).$$

Assume now that $\mathcal{L}_{\mathcal{D}}(h_i) \geq \left((10/7)c''(d\gamma^{-2} + \ln(1/\delta)) + 23\ln(9/\delta)\right)/(m/4) \geq 23\ln(9/\delta)/|A_i|$.

Using that $h_i$ and $A_i$ are independent, it follows by a Chernoff bound that

$$\Pr\left[N_i \geq (7/10)\mathcal{L}_{\mathcal{D}}(h_i)|A_i|\right] \geq 1 - \exp\left(-(3/10)^2 \mathcal{L}_{\mathcal{D}}(h_i)|A_i|/2\right)$$

$$\geq 1 - \exp\left(-(3/10)^2 \cdot 23\ln(9/\delta)/2\right)$$

$$> 1 - \delta/9.$$

Thus there is an event $E_i''$ of probability at least $1 - \delta/9$, on which, if

$$\mathcal{L}_{\mathcal{D}}(h_i) \geq \frac{(10/7)c''(d\gamma^{-2} + \ln(1/\delta)) + 23\ln(9/\delta)}{m/4}$$

then

$$N_i \geq (7/10)\mathcal{L}_{\mathcal{D}}(h_i)\,|A_i|$$
$$= (7/10)\mathcal{L}_{\mathcal{D}}(h_i)\,m/4$$
$$\geq c''(d\gamma^{-2} + \ln(1/\delta)).$$

Combining it all, we have that on the event $E_i \cap E_i' \cap E_i''$, which occurs with probability at least $1 - \delta/3$, if $\mathcal{L}_{\mathcal{D}}(h_i) \geq \left((10/7)c''(d\gamma^{-2} + \ln(1/\delta)) + 23\ln(9/\delta)\right)/(m/4)$, then every $h = \mathcal{A}_\nu^*(C_k)$ for a $C_k \in $ *Sub-Sample*$(A_0, B_j)$ with $j \neq i$ has:

$$\Pr_{x \sim \mathcal{D}} \left[ h_i(x) \neq c(x) \wedge h(x) \neq c(x) \right] \leq \tfrac{1}{200}\mathcal{L}_{\mathcal{D}}(h_i)$$

By Eq. (4), this is at most

$$\Pr_{x \sim \mathcal{D}} \left[ h_i(x) \neq c(x) \wedge h(x) \neq c(x) \right] \leq \frac{1}{200} \cdot \frac{12c'}{m}\left(d\gamma^{-2} + \ln(1/\delta)\right)$$

$$\leq \frac{c'}{16m}\left(d\gamma^{-2} + \ln(1/\delta)\right).$$

On the other hand, if $\mathcal{L}_{\mathcal{D}}(h_i) < \left(c''(d\gamma^{-2} + \ln(1/\delta)) + 23\ln(9/\delta)\right)/(m/4)$, then

$$\Pr_{x \sim \mathcal{D}} \left[ h_i(x) \neq c(x) \wedge h(x) \neq c(x) \right] \leq \mathcal{L}_{\mathcal{D}}(h_i)$$

$$\leq \left(c''(d\gamma^{-2} + \ln(1/\delta)) + 23\ln(9/\delta)\right)/(m/4)$$

$$\leq 4c''(d\gamma^{-2} + 24\ln(1/\delta) + 23\ln 9)/m$$

Using that $23 \cdot \ln 9 < 51 \leq 51 d\gamma^{-2}$, the above is at most $204c''(d\gamma^{-2} + \ln(1/\delta))/m$. Fixing the constant $c'$ to $c' \geq (16 \cdot 204)c''$, this is at most

$$\frac{c'}{16m}\left(d\gamma^{-2} + \ln(1/\delta)\right).$$

We conclude that on the event $\bigcap_{i=1,2,3}\{E_i \cap E_i' \cap E_i''\}$, which occurs with probability at least $1 - \delta$ by a union bound, it holds for all $i$ and all $C_k \in $ Sub-Sample$(A_0, B_j)$ with $j \neq i$ that the hypothesis $h = \mathcal{A}_\nu^*(C_k)$ satisfies:

$$\Pr_{x \sim \mathcal{D}} \left[ h_i(x) \neq c(x) \wedge h(x) \neq c(x) \right] \leq \frac{c'}{16m}\left(d\gamma^{-2} + \ln(1/\delta)\right).$$

Now consider an $x$ on which $\hat{h}_{m,B}$ errs. On such an $x$, the majority among the classifiers

$$\bigcup_{C_i \in \text{Sub-Sample}(S_{1:m},B)} \{\mathcal{A}_\nu^*(C_i)\} \;=\; \bigcup_{i=1,2,3} \;\bigcup_{C_k \in \text{Sub-Sample}(S_{1:m/4},B_i)} \{\mathcal{A}_\nu^*(C_k)\}$$

errs. For the majority to err, there must be an $i \in \{1,2,3\}$ for which the majority of

$$\bigcup_{C_k \in \text{Sub-Sample}(S_{1:m/4},B_i)} \{\mathcal{A}_\nu^*(C_k)\}$$

errs. This is equivalent to $h_i(x) \neq c(x)$. Furthermore, even when all of the classifiers in

$$\bigcup_{C_k \in \text{Sub-Sample}(S_{1:m/4},B_i)} \{\mathcal{A}_\nu^*(C_k)\}$$

err, there still must be another $(1/6)$-fraction of all the classifiers

$$\bigcup_{i=1,2,3} \;\bigcup_{C_k \in \text{Sub-Sample}(S_{1:m/4},B_i)} \{\mathcal{A}_\nu^*(C_k)\}$$

that err. This follows since each of the three recursive calls in *Sub-Sample* generated equally many classifiers/samples. It follows that if we pick a uniform random $i \in \{1,2,3\}$ and a uniform random hypothesis $h$ in

$$\bigcup_{j \in \{1,2,3\}\setminus\{i\}} \;\bigcup_{C_k \in \text{Sub-Sample}(S_{1:m/4},B_j)} \{\mathcal{A}_\nu^*(C_k)\}$$

then with probability at least $(1/3)(1/6)(3/2) = 1/12$, we have that $h_i(x) \neq c(x) \wedge h(x) \neq c(x)$. It follows by linearity of expectation that on the event $\bigcap_{i=1,2,3}\{E_i \cap E_i' \cap E_i''\}$, we have:

$$\mathcal{L}_{\mathcal{D}}(\hat{h}_{m,B}) \;\leq\; 12 \cdot \frac{c'}{16m}\left(d\gamma^{-2} + \ln(1/\delta)\right) \;<\; \frac{c'}{m}\left(d\gamma^{-2} + \ln(1/\delta)\right).$$

This completes the inductive proof and shows Theorem A.1.

# B  Lower bound

In this section, we prove the following lower bound which directly implies Theorem 2 from the main paper:

**Theorem B.1.** *There is a universal constant $\alpha > 0$ such that for all integers $d \in \mathbb{N}$ and every $2^{-d} < \gamma < 1/80$, there is a finite set $X$, a concept class $C \subset X \to \{-1,1\}$ and a hypothesis set $\mathcal{H} \subseteq X \to \{-1,1\}$ of VC-dimension at most $d$, such that for every integer $m \in \mathbb{N}$ and $0 < \delta < 1/3$, there is a distribution $\mathcal{D}$ over $X$ such that the following holds:*

1. *For every $c \in C$ and every distribution $\mathcal{D}'$ over $X$, there is an $h \in \mathcal{H}$ with*

$$\Pr_{x \sim \mathcal{D}'}\left[h(x) \neq c(x)\right] \;\leq\; 1/2 - \gamma.$$

2. *For any algorithm $\mathcal{A}$, there is a concept $c \in C$ such that with probability at least $\delta$ over a set of $m$ samples $S \sim \mathcal{D}^m$, the classifier $\mathcal{A}(S) \in X \to \{-1,1\}$ produced by $\mathcal{A}$ on $S$ and $c(S)$ must have*

$$\mathcal{L}_{\mathcal{D}}(\mathcal{A}(S)) \;\geq\; \alpha \cdot \frac{d\gamma^{-2} + \ln(1/\delta)}{m}.$$

Theorem B.1 immediately implies Theorem 2 by solving the equation in the second statement for $\varepsilon = \mathcal{L}_{\mathcal{D}}(\mathcal{A}(S))$.

The proof of the term $\ln(1/\delta)/m$ in the lower bound follows from previous work. In particular, we could let $C = \mathcal{H}$ and invoke the tight lower bounds for PAC-learning in the realizable setting [5]. Thus, we focus on $\delta = 1/3$ and only need to prove that the loss of $\mathcal{A}(S)$ is at least $\alpha d\gamma^{-2}/m$ with probability $1/3$ over $S$.

For the proof, we make use of the following lemma by Grønlund et al. [9] to construct the 'hard' hypothesis set $\mathcal{H}$ and concept class $C$:

**Lemma B.1** (Grønlund et al. [9]). *For every $\gamma \in (0, 1/40), \delta \in (0, 1)$ and integers $k \leq u$, there exists a distribution $\mu = \mu(u, d, \gamma, \delta)$ over a hypothesis set $\mathcal{H} \subset \mathcal{X} \to \{-1, 1\}$, where $\mathcal{X}$ is a set of size $u$, such that the following holds.*

1. *For all $\mathcal{H} \in \mathrm{supp}(\mu)$, we have $|\mathcal{H}| = N$; and*

2. *For every labeling $\ell \in \{-1, 1\}^u$, if no more than $k$ points $x \in \mathcal{X}$ satisfy $\ell(x) = -1$, then*

$$\Pr_{\mathcal{H} \sim \mu} \left[ \exists f \in \Delta(\mathcal{H}) : \forall x \in \mathcal{X} : \ell(x) f(x) \geq \gamma \right] \geq 1 - \delta.$$

*where $N = \Theta\big(\gamma^{-2} \ln u \ln(\gamma^{-2} \ln u \delta^{-1}) e^{\Theta(\gamma^2 k)}\big)$.*

To prove Theorem B.1 for a given $\gamma \in (2^{-d}, 1/80)$ and $m, d \in \mathbb{N}$, let $u = k$ for a $u$ to be determined. Invoke Lemma B.1 with $\delta = 1/2$ and $\gamma' = 2\gamma$ to conclude that there exists a hypothesis set $\mathcal{H}$ such that among all labelings $\ell \in \{-1, 1\}^u$, at least half of them satisfy:

$$\exists f \in \Delta(\mathcal{H}) : \forall x \in \mathcal{X} : \ell(x) f(x) \geq 2\gamma.$$

Moreover, we have $N = |\mathcal{H}| = \Theta\big(\gamma^{-2} \ln u \ln(\gamma^{-2} \ln u) e^{\Theta(\gamma^2 u)}\big)$. Let the concept class $C$ be the set of such labelings.

For the given VC-dimension $d$, we need to bound the VC-dimension of $\mathcal{H}$ by $d$. For this, note that the VC-dimension is bounded by $\lg |\mathcal{H}| = \Theta(\gamma^2 u) + \lg(\gamma^{-2} \lg u))$. Using that $\gamma \geq 2^{-d}$, this is at most $\Theta(\gamma^2 u + d + \lg \lg u)$. We thus choose $u = \Theta(\gamma^{-2} d)$ which implies the claimed VC-dimension of $\mathcal{H}$.

Next, we have to argue that any concept $c \in C$ can be $\gamma$-weakly learned from $\mathcal{H}$. That is, the first statement of Theorem B.1 holds for $\mathcal{H}, C$. To see this, we must show that for every distribution $\mathcal{D}$ over $\mathcal{X}$, there is a hypothesis $h \in \mathcal{H}$ such that $\Pr_{x \sim \mathcal{D}}[h(x) = c(x)] \geq 1/2 + \gamma$. To argue that this is indeed the case, let $f \in \Delta(\mathcal{H})$ satisfy $\forall x \in \mathcal{X} : c(x) f(x) \geq 2\gamma$. Such an $f$ exists by definition of $C$. Then, $\mathbb{E}_{x \sim \mathcal{D}}[c(x) f(x)] \geq 2\gamma$. Since $f(x)$ is a convex combination of hypotheses from $\mathcal{H}$, it follows that there is a hypothesis $h \in \mathcal{H}$ also satisfying $\mathbb{E}_{x \sim \mathcal{D}}[c(x) h(x)] \geq 2\gamma$. But

$$
\begin{aligned}
\mathbb{E}_{x \sim \mathcal{D}} [c(x) h(x)] &= \sum_{x \in \mathcal{X}} \mathcal{D}(x) c(x) h(x) \\
&= \sum_{x \in \mathcal{X} : c(x) = h(x)} \mathcal{D}(x) - \sum_{x \in \mathcal{X} : c(x) \neq h(x)} \mathcal{D}(x) \\
&= \Pr_{x \sim \mathcal{D}} [c(x) = h(x)] - \Pr_{x \sim \mathcal{D}} [c(x) \neq h(x)] \\
&= \Pr_{x \sim \mathcal{D}} [c(x) = h(x)] - (1 - \Pr_{x \sim \mathcal{D}} [c(x) = h(x)]) \\
&= 2 \Pr_{x \sim \mathcal{D}} [c(x) = h(x)] - 1.
\end{aligned}
$$

Hence, $2 \cdot \Pr_{x \sim \mathcal{D}}[c(x) = h(x)] - 1 \geq 2\gamma \implies \Pr_{x \sim \mathcal{D}}[c(x) = h(x)] \geq 1/2 + \gamma$ as claimed.

We have thus constructed $\mathcal{H}$ and $C$ satisfying the first statement of Theorem B.1, where $C$ contains at least half of all possible labelings of the points $\mathcal{X} = \{x_1, \ldots, x_u\}$ with $u = \Theta(\gamma^{-2} d)$. For the remainder of the proof, we assume $u$ is at least some large constant, which is true for $\gamma$ small enough.

What remains is to establish the second statement of Theorem B.1. For this, we first define the hard distribution $\mathcal{D}$ over $\mathcal{X}$. The distribution $\mathcal{D}$ returns the point $x_1$ with probability $1 - (u - 1)/4m$ and with the remaining probability $(u - 1)/4m$ it returns a uniform random sample $x_i$ among $x_2, \ldots, x_u$. Also, let $c$ be a uniform random concept drawn from $C$.

Let $\mathcal{A}$ be any (possibly randomized) learning algorithm that on a set of samples $S$ from $\mathcal{X}$ and a labeling $\ell(S)$ of $S$ that is consistent with at least one concept $c \in C$ (i.e. $\ell(S) = c(S)$), outputs a hypothesis $h_{S, \ell(S)}$ in $\mathcal{X} \to \{-1, 1\}$. The algorithm $\mathcal{A}$ is not constrained to output a hypothesis from $\Delta(\mathcal{H})$ or $\mathcal{H}$, but instead may output any desirable hypothesis in $\mathcal{X} \to \{-1, 1\}$, using the full knowledge of $C, \ell(S), \mathcal{H}$ and the promise that $c \in C$. Our goal is to show that

$$\mathbb{E}_{c \sim C} \left[ \Pr_{S \sim \mathcal{D}^m} \left[ \Pr_{x \sim \mathcal{D}} [h_{S, c(S)}(x) \neq c(x)] \geq \alpha' \frac{d\gamma^{-2}}{m} \right] \right] \geq 1/3 \tag{5}$$

where $c \sim C$ denotes the uniform random choice of $c$. Notice that if this is the case, there must exist a concept $c$ for which

$$\Pr_{S \sim \mathcal{D}^m} \left[ \Pr_{x \sim \mathcal{D}} [h_{S,c(S)}(x) \neq c(x)] \geq \alpha' \frac{d\gamma^{-2}}{m} \right] \geq 1/3.$$

To establish Eq. (5), we start by observing that for any randomized algorithm $\mathcal{A}$, there is a deterministic algorithm $\mathcal{A}'$ obtaining a smaller than or equal value of the left hand side of Eq. (5) (by Yao's principle). Thus, we assume from here on that $\mathcal{A}$ is deterministic.

The main idea in our proof is to first show that conditioned on the set $S$ and label $c(S)$, the concept $c$ is still largely unknown. We formally measure this by arguing that the binary Shannon entropy of $c$ is large conditioned on $S$ and $c(S)$. Next, we argue that if a learning algorithm often manages to produce an accurate hypothesis from $S$ and $c(S)$, then that reveals a lot of information about $c$, i.e. the entropy of $c$ is small conditioned on $S$ and $c(S)$. This contradicts the first statement and thus the algorithm cannot produce an accurate hypothesis. We now proceed with the two steps.

**Large Conditional Entropy.** Consider the binary Shannon entropy of the uniform random $c$ conditioned on $S$ and $c(S)$, denoted $H(c \mid S, c(S))$. We know that $H(c) = \lg |C| \geq \lg(2^u/2) = u - 1$. The random variable $c$ is independent of $S$, hence $H(c \mid S) = H(c)$. We therefore have $H(c \mid S, c(S)) \geq H(c \mid S) - H(c(S) \mid S) = u - 1 - H(c(S) \mid S)$. For a fixed $s \in \mathcal{X}^m$, let $p_s = \Pr_{S \sim \mathcal{D}^m}[S = s]$. Then $H(c(S) \mid S) = \sum_{s \in \mathcal{X}^m} p_s H(c(S) \mid S = s) \leq \sum_{s \in \mathcal{X}^m} p_s |s|$, where the last step follows from the fact that, conditioned on $s$, the labeling $c(s)$ consists of $|s|$ signs. Note that the size of the set $|s|$ is possibly smaller than $m$ due to repetitions.

Now notice that $\Pr[|S| > u/3]$ is exponentially small in $u$ since each of the $m$ samples from $\mathcal{D}$ is among $x_2, \ldots, x_u$ with probability only $(u - 1)/(4m)$. Therefore, we get $H(c(S) \mid S) \leq u/3 + \exp(-\Omega(u))u \leq u/2 - 1$. It follows that

$$H(c \mid S, c(S)) \geq u - 1 - (u/2 - 1) = u/2. \tag{6}$$

**Accuracy Implies Low Entropy.** Now assume that $h_{S,c(S)}$ is such that $\Pr_{x \sim \mathcal{D}}[h_{S,c(S)} \neq c(x)] < \alpha' d\gamma^{-2}/m$ for a sufficiently small constant $\alpha'$. Any point $x_i$ where $c(x_i)$ disagrees with $h_{S,c(S)}(x_i)$ adds at least $1/(4m)$ to $\Pr_{x \sim \mathcal{D}}[h_{S,c(S)} \neq c(x)]$ (the point $x_1$ would add more), hence $h_{S,c(S)}$ makes a mistake on at most $\alpha' d\gamma^{-2}/m \cdot (4m) = 4\alpha' d\gamma^{-2}$ points. Recalling that $u = \Theta(d\gamma^{-2})$, we get that for $\alpha'$ small enough, this is less than $u/100$. Thus, conditioned on $\Pr_{x \sim \mathcal{D}}[h_{S,c(S)} \neq c(x)] < \alpha' d\gamma^{-2}/m$ and $h_{S,c(S)}$, we get that the entropy of the concept $c$ is no more than $\lg\left(\sum_{i=0}^{u/100} \binom{u}{i}\right)$ since $c$ is within a Hamming ball of radius $u/100$ from $h_{S,c(S)}$. Now $\sum_{i=0}^{u/100} \binom{u}{i} \leq 2^{H_b(1/100)u}$, where $H_b$ is the binary entropy of a Bernoulli random variable with success probability $1/100$. Numerical calculations give $H_b(1/100) = (1/100)\lg_2(100) + (99/100)\lg_2(100/99) < 0.09$. Thus

$$H\left(c \mid h_{S,c(S)}, \Pr_{x \sim \mathcal{D}}[h_{S,c(S)} \neq c(x)] < \alpha' d\gamma^{-2}/m\right) \leq 0.09u. \tag{7}$$

Now let $X_{S,c}$ be an indicator random variable for the event that $\Pr_{x \sim \mathcal{D}}[h_{S,c(S)} \neq c(x)] < \alpha' d\gamma^{-2}/m$. Then $H(c \mid S, c(S)) \leq H(c \mid S, c(S), h_{S,c(S)}, X_{S,c}) + H(X_{S,c})$. Here we remark that we add $h_{S,c(S)}$ in the conditioning for free since it depends only on $S$ and $c(S)$. Adding $X_{S,c}$ costs at most its entropy which satisfies $H(X_{S,c}) \leq 1$. Since removing variables that we condition on only increases entropy, we get $H(c \mid S, c(S)) \leq H(c \mid h_{S,c(S)}, X_{S,c}) + 1$. Now observe that $H(c \mid h_{S,c(S)}, X_{S,c}) = \Pr[X_{S,c} = 1]H(c \mid h_{S,c(S)}, X_{S,c} = 1) + \Pr[X_{S,c} = 0]H(c \mid h_{S,c(S)}, X_{S,c} = 0)$. The latter entropy we simply bound by $u$ and the former is bounded by $0.09u$ by Eq. (7). Thus $H(c \mid S, c(S)) \leq 1 + \Pr[X_{S,c} = 1]0.09u + (1 - \Pr[X_{S,c} = 1])u$.

**Combining the Bounds.** Combining the above with Eq. (6) we conclude that

$$1 + \Pr[X_{S,c} = 1]0.09u + (1 - \Pr[X_{S,c} = 1])u \geq u/2.$$

It follows that $\Pr[X_{S,c} = 1] \leq 2/3$. This completes the proof since

$$\mathbb{E}_{c \sim C} \left[ \Pr_{S \sim \mathcal{D}^m} \left[ \Pr_{x \sim \mathcal{D}} [h_{S,c(S)}(x) \neq c(x)] \geq \alpha' \frac{d\gamma^{-2}}{m} \right] \right] = \mathbb{E}_{c \sim C} \left[ \mathbb{E}_{S \sim \mathcal{D}^m} [(1 - X_{S,c})] \right] = 1 - \Pr[X_{S,c} = 1]$$

and thus

$$\mathop{\mathbb{E}}_{c\sim C}\left[\mathop{\mathrm{Pr}}_{S\sim\mathcal{D}^m}\left[\mathop{\mathrm{Pr}}_{x\sim\mathcal{D}}[h_{S,c(S)}(x)\neq c(x)]\geq\alpha'\frac{d\gamma^{-2}}{m}\right]\right]\;\geq\;\frac{1}{3}.$$

This finishes the proof of Theorem B.1.

## C Proofs for the margin-based generalization bound for voting classifiers

This section covers the proofs of all lemmas needed to show the generalization bound for voting classifiers with large margins (Theorem 4 in the main paper) that did not fit into the main text.

### C.1 Proofs of key properties of $\mathcal{D}_{f,t}$

First, we present the proofs of Lemma 1, 2, and 3 from the main paper covering different properties of the distribution $\mathcal{D}_{f,t}$.

**Restatement of Lemma 1.** *For any $x\in X$, any $f\in\Delta(\mathcal{H})$ and any $\mu>0$:*

$$\mathop{\mathrm{Pr}}_{g\sim\mathcal{D}_{f,t}}\left[|f(x)-g(x)|\geq\mu\right]\;<\;5\exp(-\mu^2 t/32).$$

*Proof.* This lemma follows using standard concentration inequalities: In the first step of sampling $g$ from $\mathcal{D}_{f,t}$, where we draw $t$ i.i.d. hypotheses, it follows from Hoeffding's inequality that the hypothesis $g'(x)=(1/t)\sum_{i=1}^t h_i'(x)$ satisfies

$$\mathop{\mathrm{Pr}}_{g'}\left[|f(x)-g'(x)|\geq\mu/2\right]\;\leq\;2\exp\left(-2(\mu/2)^2 t^2/(4t)\right)\;=\;2\exp(-\mu^2 t/8).$$

In the second step, we first get by a Chernoff bound that $\mathrm{Pr}[t'<t/4]<\exp(-t/16)$. Secondly, let us condition on any fixed value of $t'$ that is at least $t/4$. Then $h_1,\ldots,h_{t'}$ is a uniform sample without replacement from $h_1',\ldots,h_t'$. It follows by a Hoeffding bound without replacement that

$$\mathrm{Pr}\left[|g(x)-g'(x)|\geq\mu/2\right]\;\leq\;2\exp\left(-2(\mu/2)^2(t')^2/(4t')\right)\;<\;2\exp(-\mu^2 t/32).$$

In total, we conclude that

$$\mathrm{Pr}\left[|f(x)-g(x)|\geq\mu\right]\;<\;2\exp(-\mu^2 t/8)+\exp(-t/16)+2\exp(-\mu^2 t/32)\;<\;5\exp(-\mu^2 t/32).\;\square$$

**Restatement of Lemma 2.** *For any $x\in X$, any $f\in\Delta(\mathcal{H})$ and any $\mu\geq 1/t$:*

$$\mathop{\mathrm{Pr}}_{g\sim\mathcal{D}_{f,t}}\left[|g(x)|\leq\mu\right]\;\leq\;2\mu\sqrt{t}.$$

*Proof.* Let $h_1',\ldots,h_t'$ be the hypotheses sampled in the first step of drawing $g$. Define $\sigma_i$ to be 1 if $h_i'$ is sampled in $g$ and $-1$ otherwise. That is, we have

$$g(x)=\frac{1}{|\{i:\sigma_i=1\}|}\sum_{i:\sigma_i=1}h_i'(x).$$

Let $\Gamma=\sum_{i=1}^t h_i'(x)$. Then

$$\Gamma+\sum_{i=1}^t\sigma_i h_i'(x)$$

$$=\sum_{i:\sigma_i=1}h_i'(x)+\sum_{i:\sigma_i=-1}h_i'(x)+\sum_{i=1}^t\sigma_i h_i'(x)$$

$$=2\sum_{i:\sigma_i=1}h_i'(x)=2t'g(x).$$

Therefore, $|g(x)|\leq\mu$ if and only if

$$\left|\frac{\Gamma+\sum_i\sigma_i h_i'(x)}{2t'}\right|\leq\mu.$$

Since $t' \le t$, this implies

$$\left| \frac{\Gamma + \sum_i \sigma_i h'_i(x)}{2t} \right| \le \mu.$$

Hence, we have $\Pr[|g(x)| \le \mu] \le \Pr[\sum_i \sigma_i h'_i(x) \in -\Gamma \pm 2t\mu]$. By Erdös' improved Littlewood-Offord lemma, as long as $2t\mu \ge 2$, this happens with probability at most $2t\mu\binom{t}{\lfloor t/2 \rfloor}2^{-t}$. The central binomial coefficient satisfies $\binom{t}{\lfloor t/2 \rfloor} \le 2^t/\sqrt{\pi t/2} \le 2^t/\sqrt{t}$ and thus the probability is at most $2t\mu/\sqrt{t} = 2\mu\sqrt{t}$. $\qquad\square$

Finally, we prove Lemma 3 from the main paper:

**Restatement of Lemma 3.** *For any distribution $\mathcal{D}$ over $X \times \{-1, 1\}$, any $t \ge 36$ and any voting classifier $f \in \Delta(\mathcal{H})$ for a hypothesis set $\mathcal{H} \subset X \rightarrow \{-1, 1\}$, we have:*

$$\mathcal{L}_{\mathcal{D}}(f) \le 3\mathcal{L}^t_{\mathcal{D}}(f).$$

For the proof, we first need the following auxiliary lemma:

**Lemma C.1.** *For any $x \in X$ and any $f \in \Delta(\mathcal{H})$, if $f(x) \ne 0$, then*

$$\Pr_{g \sim \mathcal{D}_{f,t}} \left[ \text{sign}(f(x)) = \text{sign}(g(x)) \right] \ge 1/2 - 1/\sqrt{t}.$$

*Proof.* If we condition on $t'$, then $h_1, \dots, h_{t'}$ are i.i.d samples from $\mathcal{D}_f$ and thus $\Pr[\text{sign}(g(x)) = \text{sign}(f(x))] \ge \Pr[\text{sign}(g(x)) = -\text{sign}(f(x))]$. We therefore have $\Pr[\text{sign}(f(x)) = \text{sign}(g(x))] \ge \Pr[g(x) \ne 0]/2$, regardless of $t'$. We thus only need to bound $\Pr[g(x) \ne 0]$. For this, Lemma 2 with $\mu = 1/t$ implies $\Pr[g(x) = 0] \le 2/\sqrt{t}$. $\qquad\square$

Using this lemma, we can prove Lemma 3:

*Proof of Lemma 3 from the main paper.* Consider any example $(x, y) \in X \times \{-1, 1\}$ for which $\Pr_{g \sim \mathcal{D}_{f,t}}[yg(x) \le 0] < 1/2 - 1/\sqrt{t}$. By Lemma C.1, it must be the case that $\text{sign}(f(x)) = y$. We therefore have by Markov's inequality:

$$
\begin{aligned}
\mathcal{L}_{\mathcal{D}}(f) &\le \Pr_{(x,y) \sim \mathcal{D}}[\Pr_{g \sim \mathcal{D}_{f,t}}[yg(x) \le 0] \ge 1/2 - 1/\sqrt{t}] \\
&\le \frac{\mathbb{E}_{(x,y) \sim \mathcal{D}}[\Pr_{g \sim \mathcal{D}_{f,t}}[yg(x) \le 0]]}{1/2 - 1/\sqrt{t}} \\
&= \mathcal{L}^t_{\mathcal{D}}(f)/(1/2 - 1/\sqrt{t}) \\
&\le 3\mathcal{L}^t_{\mathcal{D}}(f). \qquad\qquad\square
\end{aligned}
$$

## C.2 Relating generalization error to the ghost set

In the following, we give the proof of Lemma 6 from the main paper:

**Restatement of Lemma 6.** *For $m \ge 2400^2$ any $t$ and any $f$, it holds that:*

$$\Pr_{S}\left[ \sup_{f \in \Delta(\mathcal{H})} |\mathcal{L}^t_S(f) - \mathcal{L}^t_{\mathcal{D}}(f)| > \tfrac{1}{1200} \right] \le 2 \cdot \Pr_{S,S'}\left[ \sup_{f \in \Delta(\mathcal{H})} |\mathcal{L}^t_S(f) - \mathcal{L}^t_{S'}(f)| > \tfrac{1}{2400} \right].$$

*Proof.* The proof uses standard techniques uneventfully. We can assume $\Pr_S[\sup_{f \in \Delta(\mathcal{H})} |\mathcal{L}^t_S(f) - \mathcal{L}^t_{\mathcal{D}}(f)| > 1/1200] > 0$, otherwise we are done. We have:

$$
\begin{aligned}
&\Pr_{S,S'}\left[ \sup_{f \in \Delta(\mathcal{H})} |\mathcal{L}^t_S(f) - \mathcal{L}^t_{S'}(f)| > \tfrac{1}{2400} \right] \\
&\ge \Pr_{S,S'}\left[ \sup_{f \in \Delta(\mathcal{H})} |\mathcal{L}^t_S(f) - \mathcal{L}^t_{S'}(f)| > \tfrac{1}{2400} \wedge \sup_{f \in \Delta(\mathcal{H})} |\mathcal{L}^t_S(f) - \mathcal{L}^t_{\mathcal{D}}(f)| > \tfrac{1}{1200} \right] \\
&= \Pr_{S}\left[ \sup_{f \in \Delta(\mathcal{H})} |\mathcal{L}^t_S(f) - \mathcal{L}^t_{\mathcal{D}}(f)| > \tfrac{1}{1200} \right] \times \\
&\quad \Pr_{S,S'}\left[ \sup_{f \in \Delta(\mathcal{H})} |\mathcal{L}^t_S(f) - \mathcal{L}^t_{S'}(f)| > \tfrac{1}{2400} \;\Big|\; \sup_{f \in \Delta(\mathcal{H})} |\mathcal{L}^t_S(f) - \mathcal{L}^t_{\mathcal{D}}(f)| > \tfrac{1}{1200} \right].
\end{aligned}
$$

Fix a data set $S$ in the non-empty event $\sup_{f\in\Delta(\mathcal{H})} |\mathcal{L}_S^t(f) - \mathcal{L}_{\mathcal{D}}^t(f)| > 1/1200$. Let $f^* \in \mathcal{H}$ be any hypothesis on which $|\mathcal{L}_S^t(f^*) - \mathcal{L}_{\mathcal{D}}^t(f^*)| > 1/1200$. The hypothesis $f^*$ does not depend on $S'$ but only on $S$. We now condition on $S$ as well and get:

$$\Pr_{S,S'}\Big[\sup_{f\in\Delta(\mathcal{H})} |\mathcal{L}_S^t(f) - \mathcal{L}_{S'}^t(f)| > \tfrac{1}{2400} \,\Big|\, S;\ \sup_{f\in\Delta(\mathcal{H})} |\mathcal{L}_S^t(f) - \mathcal{L}_{\mathcal{D}}^t(f)| > \tfrac{1}{1200}\Big]$$

$$\geq \Pr_{S'}\Big[|\mathcal{L}_S^t(f^*) - \mathcal{L}_{S'}^t(f^*)| > \tfrac{1}{2400} \,\Big|\, S;\ \sup_{f\in\Delta(\mathcal{H})} |\mathcal{L}_S^t(f) - \mathcal{L}_{\mathcal{D}}^t(f)| > \tfrac{1}{1200}\Big]$$

$$\geq \Pr_{S'}\Big[|\mathcal{L}_{S'}^t(f^*) - \mathcal{L}_{\mathcal{D}}^t(f^*)| \leq \tfrac{1}{2400} \,\Big|\, S;\ \sup_{f\in\Delta(\mathcal{H})} |\mathcal{L}_S^t(f) - \mathcal{L}_{\mathcal{D}}^t(f)| > \tfrac{1}{1200}\Big].$$

Here the last inequality follows because the events $|\mathcal{L}_{S'}^t(f^*) - \mathcal{L}_{\mathcal{D}}^t(f^*)| \leq 1/2400$ and $|\mathcal{L}_S^t(f^*) - \mathcal{L}_{\mathcal{D}}^t(f^*)| > 1/1200$ (which holds by definition of $f^*$) implies $|\mathcal{L}_S^t(f^*) - \mathcal{L}_{S'}^t(f^*)| > 1/2400$. Since $f^*$ is fixed and independent of $S'$, we may now use Hoeffding's inequality to conclude

$$\Pr_{S'}\Big[|\mathcal{L}_{S'}^t(f^*) - \mathcal{L}_{\mathcal{D}}^t(f^*)| \leq 1/2400 \,\Big|\, S;\ \sup_{f\in\Delta(\mathcal{H})} |\mathcal{L}_S^t(f) - \mathcal{L}_{\mathcal{D}}^t(f)| > 1/1200\Big] \geq 1 - 2e^{-2(1/2400)^2 m}.$$

For $m \geq 2400^2$, this is at least $1 - 2e^{-2} \geq 1/2$.

Multiplying with $\Pr[S \mid \sup_{f\in\Delta(\mathcal{H})} |\mathcal{L}_S^t(f) - \mathcal{L}_{\mathcal{D}}^t(f)| > 1/1200]$ and integrating over $S$, we get

$$\int_S \Big(\Pr_{S'}\Big[\sup_{f\in\Delta(\mathcal{H})} |\mathcal{L}_S^t(f) - \mathcal{L}_{S'}^t(f)| > \tfrac{1}{2400} \,\Big|\, S;\ \sup_{f\in\Delta(\mathcal{H})} |\mathcal{L}_S^t(f) - \mathcal{L}_{\mathcal{D}}^t(f)| > \tfrac{1}{1200}\Big]$$

$$\times \Pr\Big[S \,\Big|\, \sup_{f\in\Delta(\mathcal{H})} |\mathcal{L}_S^t(f) - \mathcal{L}_{\mathcal{D}}^t(f)| > \tfrac{1}{1200}\Big]\Big)$$

$$\geq \int_S \tfrac{1}{2} \Pr\Big[S \,\Big|\, \sup_{f\in\Delta(\mathcal{H})} |\mathcal{L}_S^t(f) - \mathcal{L}_{\mathcal{D}}^t(f)| > \tfrac{1}{1200}\Big].$$

The right hand side is simply $1/2$ and the left hand side is $\Pr_{S,S'}[\sup_{f\in\Delta(\mathcal{H})} |\mathcal{L}_S^t(f) - \mathcal{L}_{S'}^t(f)| > 1/2400 \mid \sup_{f\in\Delta(\mathcal{H})} |\mathcal{L}_S^t(f) - \mathcal{L}_{\mathcal{D}}^t(f)| > 1/1200]$. We finally conclude that for $m \geq 2400^2$, we have:

$$\Pr_{S,S'}\Big[\sup_{f\in\Delta(\mathcal{H})} |\mathcal{L}_S^t(f) - \mathcal{L}_{S'}^t(f)| > \tfrac{1}{2400}\Big] \geq \tfrac{1}{2} \Pr_S\Big[\sup_{f\in\Delta(\mathcal{H})} |\mathcal{L}_S^t(f) - \mathcal{L}_{\mathcal{D}}^t(f)| > \tfrac{1}{1200}\Big]. \qquad \square$$

### C.3  Relation to the growth function

Last, we prove Lemma 8 from the main paper, which is restated here for convenience:

**Restatement of Lemma 8.** *For any $0 < \delta < 1$, every $t$, and every $\mu \leq \delta/(9600\sqrt{t})$, we have*

$$\Pr_{S,S'}\Big[\sup_{f\in\Delta(\mathcal{H})} |\mathcal{L}_S^t(f) - \mathcal{L}_{S'}^t(f)| > \tfrac{1}{2400}\Big] \leq \sup_P 2\big|\hat{\Delta}_\delta^\mu(P)\big| \exp\big(-2m/9600^2\big).$$

*Proof.* Let $\mu \leq \delta/(9600\sqrt{t})$. We have that:

$$\Pr_{S,S'}\Big[\sup_{f\in\Delta(\mathcal{H})} |\mathcal{L}_S^t(f) - \mathcal{L}_{S'}^t(f)| > \tfrac{1}{2400}\Big]$$

$$= \int_P \Pr[P] \Pr_{S,S'}\Big[\sup_{f\in\Delta(\mathcal{H})} |\mathcal{L}_S^t(f) - \mathcal{L}_{S'}^t(f)| > \tfrac{1}{2400} \,\Big|\, P\Big]$$

$$\leq \sup_P \Pr_{S,S'}\Big[\sup_{f\in\Delta(\mathcal{H})} |\mathcal{L}_S^t(f) - \mathcal{L}_{S'}^t(f)| > \tfrac{1}{2400} \,\Big|\, P\Big]$$

$$= \sup_P \Pr_{S,S'}\Big[\sup_{f\in\Delta(\mathcal{H})} \big|\Pr_{(x,y)\sim S, g\sim\mathcal{D}_{f,t}}[yg(x) \leq 0] - \Pr_{(x,y)\sim S', g\sim\mathcal{D}_{f,t}}[yg(x) \leq 0]\big| > \tfrac{1}{2400} \,\Big|\, P\Big]$$

$$= \sup_P \Pr_{S,S'}\Big[\sup_{f\in\Delta(\mathcal{H})} \Big|\int_g \Pr[g]\Big(\Pr_{(x,y)\sim S}[yg(x) \leq 0] - \Pr_{(x,y)\sim S'}[yg(x) \leq 0]\Big)\Big| > \tfrac{1}{2400} \,\Big|\, P\Big].$$

We always have $\left(\Pr_{(x,y)\sim S}[yg(x) \leq 0] - \Pr_{(x,y)\sim S'}[yg(t) \leq 0]\right) \leq 1$, and by Lemma 13, we have $\Pr[g \notin \Delta_\delta^\mu(\mathcal{H}, P)] \leq 1/4800$, hence

$$\left| \int_g \Pr[g] \left( \Pr_{(x,y)\sim S}[yg(x) \leq 0] - \Pr_{(x,y)\sim S'}[yg(x) \leq 0] \right) \right|$$

$$\leq \Pr_{g\sim\mathcal{D}_{f,g}}\left[ g \notin \Delta_\delta^\mu(\mathcal{H}, P) \right] + \sup_{g\in\Delta_\delta^\mu(\mathcal{H},P)} \left| \Pr_{(x,y)\sim S}[yg(x) \leq 0] - \Pr_{(x,y)\sim S'}[yg(x) \leq 0] \right|$$

$$\leq \tfrac{1}{4800} + \sup_{g\in\Delta_\delta^\mu(\mathcal{H},P)} \left| \Pr_{(x,y)\sim S}[yg(x) \leq 0] - \Pr_{(x,y)\sim S'}[yg(x) \leq 0] \right|.$$

We thus have

$$\Pr_{S,S'}\left[ \sup_{f\in\Delta(\mathcal{H})} |\mathcal{L}_S^t(f) - \mathcal{L}_{S'}^t(f)| > \tfrac{1}{2400} \right]$$

$$\leq \sup_P \Pr_{S,S'}\left[ \tfrac{1}{4800} + \sup_{g\in\Delta_\delta^\mu(\mathcal{H},P)} \left| \Pr_{(x,y)\sim S}[yg(x) \leq 0] - \Pr_{(x,y)\sim S'}[yg(x) \leq 0] \right| > \tfrac{1}{2400} \,\Big|\, P \right]$$

$$= \sup_P \Pr_{S,S'}\left[ \sup_{g\in\Delta_\delta^\mu(\mathcal{H},P)} \left| \Pr_{(x,y)\sim S}[yg(x) \leq 0] - \Pr_{(x,y)\sim S'}[yg(x) \leq 0] \right| > \tfrac{1}{4800} \,\Big|\, P \right].$$

To bound this, let $\hat{\Delta}_\delta^\mu(P) = \mathrm{sign}(\Delta_\delta^\mu(\mathcal{H}, P))$. Then the above equals:

$$\sup_P \Pr_{S,S'}\left[ \sup_{h\in\hat{\Delta}_\delta^\mu(P)} \left| \Pr_{(x,y)\sim S}[h(x) \neq y] - \Pr_{(x,y)\sim S'}[h(x) \neq y] \right| > \tfrac{1}{4800} \,\Big|\, P \right].$$

Since we have restricted to the fixed set $P$, the set $\hat{\Delta}_\delta^\mu(P)$ is finite. Hence we may use the union bound to bound the above by

$$\sup_P |\hat{\Delta}_\delta^\mu(P)| \sup_{h\in\hat{\Delta}_\delta^\mu(P)} \Pr_{S,S'}\left[ \left| \Pr_{(x,y)\sim S}[h(x) \neq y] - \Pr_{(x,y)\sim S'}[h(x) \neq y] \right| > \tfrac{1}{4800} \,\Big|\, P \right].$$

For a set $P$ and hypothesis $h \in \hat{\Delta}_\delta^\mu(P)$, let $p$ denote the fraction of samples $(x, y) \in P$ for which $h(x) \neq y$. Recall that $S$ and the ghost set $S'$ are obtained from $P$ by letting $S$ be a uniform set of $m$ samples from $P$ without replacement, and $S'$ are the remaining $m$ samples. For shorthand, define $p_S = \Pr_{(x,y)\sim S}[h(x) \neq y \mid P]$ and $p_{S'}$ symmetrically. Then $p = (1/2)(p_S + p_{S'})$. By Hoeffding's inequality for sampling without replacement, we have $\Pr_{S,S'}[|p_S - p| > \varepsilon \mid P] = \Pr_S[|p_S - p| > \varepsilon \mid P] < 2\exp(-2\varepsilon^2 m)$. Setting $\varepsilon = 1/9600$, we get that for $|p - p_S| \leq 1/9600$, it must be the case that $p_S' = 2p - p_S \in p \pm 1/9600$. Hence $|p_S - p_{S'}| \leq 1/4800$ and we conclude $\Pr_{S,S'}[|p_S - p_{S'}| > 1/4800 \mid P] < 2\exp(-2m/9600^2)$. Thus, we end up with the bound

$$\Pr_{S,S'}\left[ \sup_{f\in\Delta(\mathcal{H})} |\mathcal{L}_S^t(f) - \mathcal{L}_{S'}^t(f)| > \tfrac{1}{2400} \right] \leq \sup_P 2|\hat{\Delta}_\delta^\mu(P)| \exp(-2m/9600^2). \qquad \square$$