# OpenReview forum: "Optimal Weak to Strong Learning"
_NeurIPS.cc/2022/Conference — NeurIPS 2022 Accept_

### Official Review · Reviewer_B3NF · 2022-07-10

**Rating:** 7
**Confidence:** 3
**Soundness:** 3 good
**Presentation:** 3 good
**Contribution:** 3 good

**Summary:**

The paper provides a new weak to strong learning algorithm that has less sample complexity compared to the best known algorithms. It also proves that the new algorithm achieves the optimal sample complexity by driving a lower bound. The optimal sample complexity of weak to strong learning algorithms was unknown before this work, but this work settles the sample complexity of the classic problem
of constructing a strong learner from a weak learner.

**Questions:**

I have a quick question regarding the definition of weak learning. In Definition 1, the confidence probability, $\delta$, is a fixed constant. However, according to Definition 7.1 of "Foundation of Machine Learning by M. Mohri", the statement must hold for any arbitrary $\delta>0$. Please give an explanation on these two different definitions.

**Limitations:**

The paper does not have any negative societal impact.

**Strengths And Weaknesses:**

- The paper solves an important open problem regarding the sample complexity of weak to strong learning algorithms by improving the sample complexity of AdaBoost by two logarithmic factors. For the theoretical analysis of the newly proposed algorithm, they also derive a new generalization bound for voting classifiers that improve the Breiman's result.

- AdaBoost is a voting classifier. They justify that a simple majority vote is not sufficient for obtaining an optimal sample complexity and an algorithm based on a majority of majorities is indeed necessary.

- The paper is well written and easy to follow.

-It is of sufficient interest to the audience of NeurIPS.

- I did not check the proof in the supplementary material but the technical proof seems to be sound.

Weaknesses:
The paper does not have any experimental results. I recommend the authors to incorporate some experiments to compare the sample complexity of their algorithm to other weak to strong learning algorithms such as AdaBoost on real data sets. They can also move some of parts of the proof to the supplementary material and instead incorporate some experimental results.

---

> ### Author Response · Authors · 2022-07-28
> **Regarding the definition of weak learning**
>
> Thank you for the review. Your question considered the subtle difference between our definition of weak learnability and the more common one e.g. found in Mohri's book.
>
> > I have a quick question regarding the definition of weak learning. In Definition 1, the confidence probability, $\delta$, is a fixed constant. However, according to Definition 7.1 of "Foundation of Machine Learning by M. Mohri", the statement must hold for any arbitrary $\delta > 0$. Please give an explanation on these two different definitions.
>
> When comparing the two definitions one can see that our Definition 1 is slightly weaker than Definition 7.1 in Mohri's book in the sense that every weak learner satisfying Definition 7.1 also satisfies our Definition 1, making our results more general. We will add a remark discussing this discrepancy.
>
> Within AdaBoost, both definitions are effectively equivalent. This holds as in the training phase of AdaBoost, the distributions $D_i$, on which the weak learner is called, are known and we can thus measure the achieved advantage of any hypothesis.
> We thus repeat the call of the weak learner until the desired advantage of $\gamma$ from the weak learner's definition is achieved. The confidence parameter $\delta$ of the weak learner therefore only affects the runtime of AdaBoost.

---

### Official Review · Reviewer_kJot · 2022-07-11

**Rating:** 8
**Confidence:** 3
**Soundness:** 4 excellent
**Presentation:** 3 good
**Contribution:** 4 excellent

**Summary:**

This paper considers the number of training examples to build a strong learner from a weak learning algorithm.
It presents a new algorithm with a bound that is two logarithmic factors lower than the state of the art.
In addition, it includes a lower bound asymptotically matching the upper bound.

The algorithm uses a sub-sampling scheme from Hanneke.  On each of the many subsamples, a boosting algorithm runs a weak learner guaranteeing edge $\gamma/2$ on the given sub-sample (where $\gamma$ is the edge of the weak learning algorithm) to produce an intermediate voted hypothesis.
These resulting intermediate hypotheses are then thresholded and majority (unweighted) voted to produce the final classifier.

A new theorem (Theorem 6) is applied to show each subsample suffices for the thresholded intermediate hypotheses to have generalization error at most 1/200 (with high probability).



**Questions:**

Are there situations where AdaBoost is known to require at least $\Omega$(expression in (1) )  examples for strong learning?
This seems implied by the page 3 line 80 comment, but it is not stated precisely.


**Limitations:**

yes

**Strengths And Weaknesses:**

It is very nice that the upper and lower bounds match asymptotically.

Although only a minor limitation, it seems like Theorem 6 requires that the weak learners be classifiers rather “confidence rated”.

---

> ### Author Response · Authors · 2022-07-28
> **Situations in which AdaBoost requires many examples**
>
> Thank you for the review. In the following we will try to shed a little more light on the question raised in the review:
>
> > Are there situations where AdaBoost is known to require at least $\Omega$(expression in (1) ) examples for strong learning? This seems implied by the page 3 line 80 comment, but it is not stated precisely.
>
> We apologize for the imprecise discussion of this issue. No, it is not known from previous work that AdaBoost actually needs at least the number of samples in (1). It is only known that it needs at most the number of samples in (1). We see that our wording on page 3 line 80 is imprecise. By requires, we merely meant that this would be sufficient. However, as a generalization lower bound in previous work shows, there are voting classifiers that have margins gamma on all training samples and yet must pay logarithmic factors in their sample complexity. It is thus conceivable that AdaBoost could do better, but we see no approach towards proving it and suspect that it is not possible. If one was to prove that AdaBoost does better than (1), one would have to argue that AdaBoost somehow cannot output the voting classifier with all margins gamma but poor generalization (the voting classifier in the previous lower bound). There seems to be nothing inherent in AdaBoost that would avoid outputting this voting classifier. We will of course update the wording on page 3 line 80.

---

### Official Review · Reviewer_Zmra · 2022-07-12

**Rating:** 6
**Confidence:** 4
**Soundness:** 3 good
**Presentation:** 3 good
**Contribution:** 3 good

**Summary:**

The paper presents a new algorithm that constructs a strong learner from a weak learner with less sample complexity than existing methods by two logarithmic factors. Matching upper and lower bounds for sample complexity shows that the algorithm is optimal up to multiplicative constants. In doing so, the algorithmic analysis uses a new generalization bound for voting classifiers with large margins. This work resolves open issues with respect to the sample complexity of the problem of constructing a strong learner from a weak learner.

**Questions:**

Is it possible to extend the work in the paper to address the few important related open theoretical problems? Or would a completely new approach be needed to address these open problems? The paper should at least discuss this.

Maybe it is just a formatting issue, but it is somewhat strange to have the first theorem listed as Theorem 3 as a result of the two preceding definitions Definition 1 and Definition 2. Can this be fixed? I do not believe this is a NeurIPS standard.


**Limitations:**

Yes

**Strengths And Weaknesses:**

The paper provides a theoretical contribution to an important problem in practice. This includes a new algorithm that constructs a strong learner from a weak learner using less training data than previous work, and showing that the algorithm is optimal up to multiplicative constants and resolves the sample complexity of the problem of constructing a strong learner from a weak learner.

The paper does not provide any empirical results that illustrate the theoretical results in practice, but this is not a major concern given the theoretical contribution of the paper. The paper leaves open a few important related theoretical problems, but the theoretical contributions are strong enough on their own.

---

> ### Author Response · Authors · 2022-07-28
> **Answers to the posed questions**
>
> Thank you for the review. In the following we try to address the two open questions raised in the review.
>
> > **Question 1** Is it possible to extend the work in the paper to address the few important related open theoretical problems? Or would a completely new approach be needed to address these open problems? The paper should at least discuss this.
>
> In the paper we stated three different open problems. For the number and size of sub-samples, this seems to require revisiting Hanneke's optimal PAC learner and improving that construction first. It is at least conceivable that a different sub-sampling strategy can be made to work but that would also be a nice advancement over state-of-the-art PAC learning in the realizable case. We do not find that the techniques we presented in this paper would help improve Hanneke's result, but suspect that an improvement to his result could be combined with our ideas to also get a better weak to strong learner.
>
> For the second open problem, we believe that a majority of majorities is actually necessary. This is supported by the previous lower bounds showing that there are simple majority voters that have large margins but poor generalization (paying a logarithmic factor). But perhaps a carefully analyzed tweak to AdaBoost could do the job. At least, previous work shows that such a result would have to analyze a concrete algorithm to exploit its specific properties and cannot merely rely on large margins.
>
> For the last open problem on a setting where the margin is not achieved everywhere, we see a faint connection to the k-th margin bound, but again the techniques we used in this paper will probably not work as the analysis carefully uses that on points where some boosters misclassify a sample (because they are left out of their training data), there are other boosters trained on that data, and these boosters have good margins on all samples. It seems tricky to prove that if these boosters have a few points they misclassify, these misclassified points cannot overlap with those also misclassified by other boosters.
>
> We will try to extend the conclusion to discuss the above things in further detail.
>
> $~$
>
> > **Question 2** Maybe it is just a formatting issue, but it is somewhat strange to have the first theorem listed as Theorem 3 as a result of the two preceding definitions Definition 1 and Definition 2. Can this be fixed? I do not believe this is a NeurIPS standard.
>
> NeurIPS, as most conferences, did not impose or suggest a styleguide regarding the numbering of math environments. We agree that it reads a little odd that the first theorem appears as Theorem 3. Since in our case all definitions, theorems, and lemmas appear in bulk, the confusion from having independent counters (all starting from 1) for each type of math environment will be minimal and we will thus adopt it.

---

> > ### Comment · Reviewer_Zmra · 2022-08-08
> > **Thank you for your response.**
> >
> > Appreciate your responses. Would definitely like to see extensions to the conclusion to discuss in more detail the connections between the approach presented in the paper and the different open problems mentioned. To what other problems might the proposed approach be applied? Would likely want to see the paper accepted, leaving my original judgement standing.

---

### Meta-Review · Area_Chair_QSFf · 2022-08-24

**Recommendation:** Accept
**Confidence:** Certain

**Metareview:**

The paper makes a nice progress on our understanding of boosting

**Award:**

No

---

### Decision · Program_Chairs · 2022-09-14

Accept